# Self-supervised contrastive learning performs non-linear system identification

**Rodrigo González Laiz**,* **Tobias Schmidt**\* **& Steffen Schneider**†

Institute of Computational Biology, Computational Health Center, Helmholtz Munich and
Munich Center for Machine Learning (MCML)

## Abstract

Self-supervised learning (SSL) approaches have brought tremendous success across many tasks and domains. It has been argued that these successes can be attributed to a link between SSL and identifiable representation learning: Temporal structure and auxiliary variables ensure that latent representations are related to the true underlying generative factors of the data. Here, we deepen this connection and show that SSL can perform system identification in latent space. We propose dynamics contrastive learning, a framework to uncover linear, switching linear and non-linear dynamics under a non-linear observation model, give theoretical guarantees and validate them empirically. Code: `github.com/dynamical-inference/dcl`

## 1 Introduction

The identification and modeling of dynamics from observational data is a long-standing problem in machine learning, engineering and science. A discrete-time dynamical system with latent variables $x$, observable variables $y$, control signal $u$, its control matrix $B$, and noise $\varepsilon, \nu$ can take the form

$$\begin{aligned}
x_{t+1} &= f(x_t) + Bu_t + \varepsilon_t \\
y_t &= g(x_t) + \nu_t.
\end{aligned} \tag{1}$$

and we aim to infer the functions $f$ and $g$ from a time-series of observations and, when available, control signals. Numerous algorithms have been developed to tackle special cases of this problem formulation, ranging from classical system identification methods (McGee & Schmidt, 1985; Chen & Billings, 1989) to recent generative models (Duncker et al., 2019; Linderman et al., 2017; Hälvä et al., 2021). Yet, it remains an open challenge to improve the generality, interpretability and efficiency of these inference techniques, especially when $f$ and $g$ are non-linear functions.

Contrastive learning (CL) and next-token prediction tasks have become important backbones of modern machine learning systems for learning from sequential data, proving highly effective for building meaningful latent representations (Baevski et al., 2022; Bommasani et al., 2021; Brown, 2020; Oord et al., 2018; LeCun, 2022; Sermanet et al., 2018; Radford et al., 2019). An emerging view is a connection between these algorithms and learning of *world models* (Ha & Schmidhuber, 2018; Assran et al., 2023; Garrido et al., 2024). However, the theoretical understanding of non-linear system identification by these sequence-learning algorithms remains limited.

In this work, we revisit and extend contrastive learning in the context of system identification. We uncover several surprising facts about its out-of-the-box effectiveness in identifying dynamics and unveil common design choices in SSL systems used in practice. Our theoretical study extends identifiability results (Hyvarinen & Morioka, 2016; 2017; Hyvarinen et al., 2019; Zimmermann et al., 2021; Roeder et al., 2021) for CL towards dynamical systems. While our theory makes several predictions about capabilities of standard CL, it also highlights shortcomings. To overcome these and enable interpretable dynamics inference across a range of data generating processes, we propose a general framework for linear and non-linear system identification with CL (Figure 1).

**Background.** An influential motivation of our work is Contrastive Predictive Coding (CPC; Oord et al., 2018). CPC can be recovered as a special case of our framework when using an RNN dynamics

---

*Equal contribution.
†Correspondence: steffen.schneider@helmholtz-munich.de

model. Related works have emerged across different modalities: wav2vec (Schneider et al., 2019), TCN (Sermanet et al., 2018) and CPCv2 (Henaff, 2020). In the field of system identification, notable approaches include the Extended Kalman Filter (EKF) (McGee & Schmidt, 1985) and NARMAX (Chen & Billings, 1989). Additionally, several works have also explored generative models for general dynamics (Duncker et al., 2019) and switching dynamics, e.g. rSLDS (Linderman et al., 2017). In the Nonlinear ICA literature, identifiable algorithms for time-series data, such as Time Contrastive Learning (TCL; Hyvarinen & Morioka, 2016) for non-stationary processes and Permutation Contrastive Learning (PCL; Hyvarinen & Morioka, 2017) for stationary data have been proposed, with recent advances like SNICA (Hälvä et al., 2021) for more generally structured data-generating processes.

In contrast to previous work, we focus on bridging time-series representation learning through contrastive learning with the identification of dynamical systems, both theoretically and empirically. Moreover, by not relying on an explicit data-generating model, our framework offers greater flexibility. We extend and discuss the connections to related work in more detail in Appendix C.

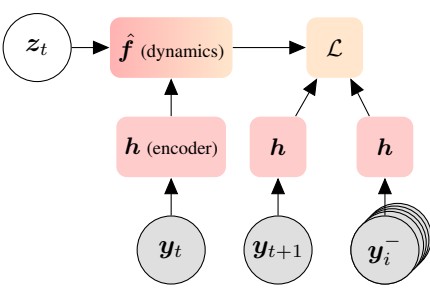

**Contributions.** We extend the existing theory on contrastive learning for time series learning and make adaptations to common inference frameworks. We introduce our CL variant (Fig. 1) in section 2, and give an identifiability result for both the latent space and the dynamics model in section 3. These theoretical results are later empirically validated. We then propose a practical way to parameterize switching linear dynamics in section 4 and demonstrate that this formulation corroborates our theory for both switching linear system dynamics and non-linear dynamics in sections 5-6.

Figure 1: DCL framework: The encoder $\boldsymbol{h}$ is shared across the reference $\boldsymbol{y}_t$, positive $\boldsymbol{y}_{t+1}$, and negative samples $\boldsymbol{y}_i^-$. A dynamics model $\hat{\boldsymbol{f}}$ forward predicts the reference. A (possibly latent) variable $\boldsymbol{z}$ can parameterize the dynamics (cf. § 4) or external control (cf. § I). The model fits the InfoNCE loss ($\mathcal{L}$).

## 2  CONTRASTIVE LEARNING FOR TIME-SERIES

In contrastive learning, we aim to model similarities between pairs of data points (Figure 1). Our full model $\psi$ is specified by the log-likelihood

$$\log p_\psi(\boldsymbol{y}|\boldsymbol{y}^+, N) = \psi(\boldsymbol{y}, \boldsymbol{y}^+) - \log \sum_{\boldsymbol{y}^- \in N \cup \{\boldsymbol{y}^+\}} \exp(\psi(\boldsymbol{y}, \boldsymbol{y}^-)). \tag{2}$$

where $\boldsymbol{y}$ is often called the reference or anchor sample, $\boldsymbol{y}^+$ is a positive sample, $\boldsymbol{y}^- \in N$ are negative examples, and $N$ is the set of negative samples. The model $\psi$ itself is parameterized as a composition of an encoder, a dynamics model, and a similarity function and will be defined further below. We fit the model by minimizing the negative log-likelihood on the time series,

$$\min_\psi \mathcal{L}[\psi] = \min_\psi \mathbb{E}_{t, t_1, \ldots, t_M \sim U(1,T)}[-\log p_\psi(\boldsymbol{y}_{t+1}|\boldsymbol{y}_t, \{\boldsymbol{y}_{t_m}\}_{m=1}^M)] \tag{3}$$

where positive examples are just adjacent points in the time-series, and $M$ negative examples are sampled uniformly across the dataset. $U(1, T)$ denotes a uniform distribution across the discrete time steps.

To attain favourable properties for identifying the latent dynamics, we carefully design the hypothesis class for $\psi$. The motivation for this particular design will become clear later. To define the full model, a composition of several functions is necessary. Recall from Eq. 1 that the dynamics model is given as $\boldsymbol{f}$ and the mixing function is $\boldsymbol{g}$. Correspondingly, our model is composed of the encoder $\boldsymbol{h} : \mathbb{R}^D \mapsto \mathbb{R}^d$ (de-mixing), the dynamics model $\hat{\boldsymbol{f}} : \mathbb{R}^d \mapsto \mathbb{R}^d$, the similarity function $\phi : \mathbb{R}^d \times \mathbb{R}^d \mapsto \mathbb{R}$ and a correction term $\alpha : \mathbb{R}^d \mapsto \mathbb{R}$. We define their composition as[1]

$$\psi(\boldsymbol{y}, \boldsymbol{y}') := \phi(\hat{\boldsymbol{f}}(\boldsymbol{h}(\boldsymbol{y})), \boldsymbol{h}(\boldsymbol{y}')) - \alpha(\boldsymbol{y}'), \tag{4}$$

and call the resulting algorithm *dynamics contrastive learning* (DCL). Intuitively, we obtain two observed samples $(\boldsymbol{y}, \boldsymbol{y}')$ which are first mapped to the latent space, $(\boldsymbol{h}(\boldsymbol{y}), \boldsymbol{h}(\boldsymbol{y}'))$. Then, the

---

[1]Note that we can equivalently write $\phi(\tilde{\boldsymbol{h}}(\boldsymbol{x})), \tilde{\boldsymbol{h}}'(\boldsymbol{x}'))$ using two asymmetric encoder functions, see additional results in Appendix D.

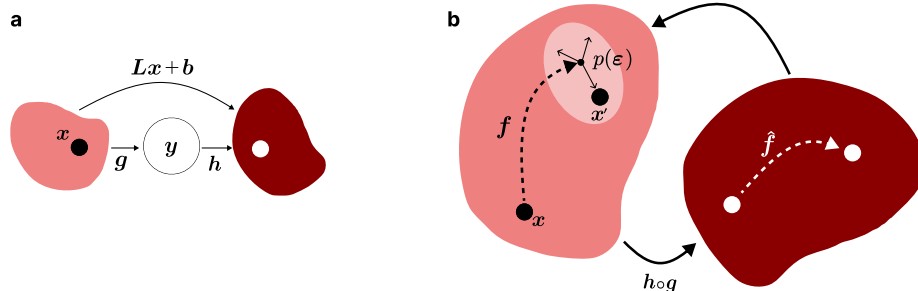

Figure 2: Graphical intuition behind Theorem 1. (a), the ground truth latent space is mapped to observables through the injective mixing function $\boldsymbol{g}$. Our model maps back into the latent space. The composition of mixing and de-mixing by the model is an affine transform. (b), dynamics in the ground-truth space are mapped to the latent space. By observing variations introduced by the system noise $\boldsymbol{\varepsilon}$, our model is able to infer the ground-truth dynamics up to an affine transform.

dynamics model is applied to $\boldsymbol{h}(\boldsymbol{y})$, and the resulting points are compared through the similarity function $\phi$. The similarity function $\phi$ will be informed by the form of (possibly induced) system noise $\boldsymbol{\varepsilon}_t$. In the simplest form, the noise can be chosen as isotropic Gaussian noise, which results in a negative squared Euclidean norm for $\phi$.

Note, the additional term $\alpha(\boldsymbol{y}')$ is a correction applied to account for non-uniform marginal distributions. It can be parameterized as a kernel density estimate (KDE) with $\log \hat{q}(\boldsymbol{h}(\boldsymbol{y}')) \approx \log q(\boldsymbol{x}')$ around the datapoints. In very special cases, the KDE makes a difference in empirical performance (App. B, Fig. 9) and is required for our theory. Yet, we found that on the time-series datasets considered, it was possible to drop this term without loss in performance (i.e., $\alpha(\boldsymbol{y}') = 0$).

## 3 STRUCTURAL IDENTIFIABILITY OF NON-LINEAR LATENT DYNAMICS

We now study the aforementioned model theoretically. The key components of our theory along with our notion of linear identifiability (Roeder et al., 2021; Khemakhem et al., 2020) are visualized in Figure 2. We are interested in two properties. First, linear identifiability of the latent space: The composition of mixing function $\boldsymbol{g}$ and model encoder $\boldsymbol{h}$ should recover the ground-truth latents up to a linear transform. Second, identifiability of the (non-linear) dynamics model: We would like to relate the estimated dynamics $\hat{\boldsymbol{f}}$ to the underlying ground-truth dynamics $\boldsymbol{f}$. This property is also called *structural identifiability* (Bellman & Åström, 1970). Our model operates on a subclass of Eq. 1 with the following properties:

**Data-generating process.** We consider a discrete-time dynamical system defined as

$$\boldsymbol{x}_{t+1} = \boldsymbol{f}(\boldsymbol{x}_t) + \boldsymbol{\varepsilon}_t, \qquad \boldsymbol{y}_t = \boldsymbol{g}(\boldsymbol{x}_t), \tag{5}$$

where $\boldsymbol{x}_t \in \mathbb{R}^d$ are latent variables, $\boldsymbol{f} : \mathbb{R}^d \mapsto \mathbb{R}^d$ is a bijective dynamics model, $\boldsymbol{\varepsilon}_t \in \mathbb{R}^d$ the system noise, and $\boldsymbol{g} : \mathbb{R}^d \mapsto \mathbb{R}^D$ is a non-linear injective mapping from latents to observables $\boldsymbol{y}_t \in \mathbb{R}^D$, $d \leq D$. We sample a total number of $T$ time steps.

We proceed by stating our main result:

**Theorem 1** (Contrastive estimation of non-linear dynamics). *Assume that*

- *(A1) A time-series dataset $\{\boldsymbol{y}_t\}_{t=1}^T$ is generated according to the ground-truth dynamical system in Eq. 5 with a bijective dynamics model $\boldsymbol{f}$ and an injective mixing function $\boldsymbol{g}$.*
- *(A2) The system noise follows an iid normal distribution, $p(\boldsymbol{\varepsilon}_t) = \mathcal{N}(\boldsymbol{\varepsilon}_t|0, \boldsymbol{\Sigma}_{\boldsymbol{\varepsilon}})$.*
- *(A3) The model $\psi$ is composed of an encoder $\boldsymbol{h}$, a dynamics model $\hat{\boldsymbol{f}}$, a correction term $\alpha$, and the similarity metric $\phi(\boldsymbol{u}, \boldsymbol{v}) = -\|\boldsymbol{u} - \boldsymbol{v}\|^2$ and attains the global minimizer of Eq. 3.*

*Then, in the limit of $T \to \infty$ for any point $\boldsymbol{x}$ in the support of the data marginal distribution:*

- *(a) The composition of mixing and de-mixing $\boldsymbol{h}(\boldsymbol{g}(\boldsymbol{x})) = \boldsymbol{L}\boldsymbol{x} + \boldsymbol{b}$ is a bijective affine transform, and $\boldsymbol{L} = \boldsymbol{Q}\boldsymbol{\Sigma}_{\epsilon}^{-1/2}$ with unknown orthogonal transform $\boldsymbol{Q} \in \mathbb{R}^{d \times d}$ and offset $\boldsymbol{b} \in \mathbb{R}^d$.*
- *(b) The estimated dynamics $\hat{\boldsymbol{f}}$ are bijective and identify the true dynamics $\boldsymbol{f}$ up to the relation $\hat{\boldsymbol{f}}(\boldsymbol{x}) = \boldsymbol{L}\boldsymbol{f}(\boldsymbol{L}^{-1}(\boldsymbol{x} - \boldsymbol{b})) + \boldsymbol{b}$.*

*Proof.* See Appendix A for the full proof, and see Fig. 2 for a graphical intuition of both results. ∎

With this main result in place, we can make statements for several systems of interest; specifically linear dynamics in latent space:

**Corollary 1.** *Contrastive learning without dynamics model, $\hat{\boldsymbol{f}}(\boldsymbol{x}) = \boldsymbol{x}$, cannot identify latent dynamics.*

In this case, even for a linear ground truth dynamics model, $\boldsymbol{f}(\boldsymbol{x}) = \boldsymbol{A}\boldsymbol{x}$, we would require that after model fitting, $\hat{\boldsymbol{f}}(\boldsymbol{x}) = \boldsymbol{x} = \boldsymbol{L}\boldsymbol{A}\boldsymbol{L}^{-1}\boldsymbol{x} + \boldsymbol{b}$, which is impossible (Theorem 1b; also see App. Eq. 22). We can fix this case by either decoupling the two encoders (Appendix D), or taking a more structured approach and parameterizing a dynamics model with a dynamics matrix:

**Corollary 2.** *For a ground-truth linear dynamical system $\boldsymbol{f}(\boldsymbol{x}) = \boldsymbol{A}\boldsymbol{x}$ and dynamics model $\hat{\boldsymbol{f}}(\boldsymbol{x}) = \hat{\boldsymbol{A}}\boldsymbol{x}$, we identify the latents up to $\boldsymbol{h}(\boldsymbol{g}(\boldsymbol{x})) = \boldsymbol{L}\boldsymbol{x} + \boldsymbol{b}$ and dynamics with $\hat{\boldsymbol{A}} = \boldsymbol{L}\boldsymbol{A}\boldsymbol{L}^{-1}$.*

This means that simultaneously fitting the system dynamics and encoding model allows us to recover the system matrix up to an indeterminacy.

**Note on Assumptions.** The required assumptions are rather practical: (A1) allows for a very broad class of dynamical systems as long as bijectivity of the *dynamics model* holds, which is the case of many systems used in the natural sciences. We consider dynamical systems with control signal $\boldsymbol{u}_t$ in Appendix I. While (A2) is a very common one in dynamical systems modeling, it can be seen more strict: We either need knowledge about the form of system noise, or inject such noise. We should note that analogous to the discussion of Zimmermann et al. (2021), it is most certainly possible to extend our results towards other classes of noise distributions by matching the log-density of $\boldsymbol{\varepsilon}$ with $\phi$. Given the common use of Normally distributed noise, however, we limited the scope of the current theory to the Normal distribution, but show vMF noise in Appendix D. (A3) mainly concerns the model setup. An apparent limitation of Def. 5 is the injectivity assumption imposed on the mixing function $\boldsymbol{g}$. In practice, a *partially observable* setting often applies, where $\boldsymbol{g}(\boldsymbol{x}) = \boldsymbol{C}\boldsymbol{x}$ maps latents into lower dimensional observations or has a lower rank than there are latent dimensions. For these systems, we can ensure injectivity through a time-lag embedding. See Appendix H for empirical validation.

## 4 ∇-SLDS: TOWARDS NON-LINEAR DYNAMICS ESTIMATION

**Piecewise linear approximation of dynamics.** Our theoretical results suggest that contrastive learning allows the fitting of non-linear bijective dynamics. This is a compelling result, but in practice it requires the use of a powerful, yet easy to parameterize dynamics model. One option is to use an RNN (Elman, 1990; Oord et al., 2018) or a Transformer (Vaswani, 2017) model to perform this link across timescales. An alternative option is to linearize the system, which we propose in the following.

$$i_t = \arg\max \|x_{t+1} - W_i x_t\|^2$$

$$MSE(x_{t+1} - W_{i_t} x_t)$$

Figure 3: The core components of the ∇-SLDS model is parameter-free, differentiable parameterization of the switching process.

We propose a new forward model for differentiable switching linear dynamics (∇-SLDS) in latent space. The estimation is outlined in Figure 3. This model allows fast estimation of switching dynamics and can be easily integrated into the DCL algorithm. The dynamics model has a trainable bank $\mathbf{W} = [\boldsymbol{W}_1, \ldots, \boldsymbol{W}_K]$ of possible dynamics matrices. $K$ is a hyperparameter. The dynamics depend on a latent variable $k_t$ and are defined as

$$\hat{\boldsymbol{f}}(\boldsymbol{x}_t; \mathbf{W}, k_t) = \boldsymbol{W}_{k_t}\boldsymbol{x}_t, \quad k_t = \mathrm{argmin}_k \|\boldsymbol{W}_k\boldsymbol{x}_t - \boldsymbol{x}_{t+1}\|^2. \quad (6)$$

Intuitively, the predictive performance of every available linear dynamical system is used to select the right dynamics with index $k_t$ from the bank $\mathbf{W}$. During training, we approximate the argmin using the Gumbel-Softmax trick (Jang et al., 2016) without hard sampling:

$$\hat{\boldsymbol{f}}(\boldsymbol{x}_t; \mathbf{W}, \boldsymbol{z}_t) = (\sum_{k=1}^{K} z_{t,k}\boldsymbol{W}_k)\boldsymbol{x}_t, \ z_{t,k} = \frac{\exp(\lambda_k/\tau)}{\sum_j \exp(\lambda_j/\tau)}, \ \lambda_k = \frac{1}{\|\boldsymbol{W}_k\boldsymbol{x}_t - \boldsymbol{x}_{t+1}\|^2} + g_k. \quad (7)$$

Note that the dynamics model $\hat{\boldsymbol{f}}(\boldsymbol{x}_t; \mathbf{W}, \boldsymbol{z}_t)$ depends on an additional latent variable $\boldsymbol{z}_t = [z_{t,1}, \ldots, z_{t,K}]^\top$ which contains probabilities to parametrize the dynamics. During inference, we

can obtain the index $k_t = \arg\max_k z_{t,k}$. The variables $g_k$ are samples from the Gumbel distribution (Jang et al., 2016) and we use a temperature $\tau$ to control the smoothness of the resulting probabilities. During pilot experiments, we found that the reciprocal parameterization of the logits outperforms other choices for computing an argmin, like flipping the sign.

**From linear switching to non-linear dynamics.** Non-linear system dynamics of the general form in Eq. 5 can be approximated using our switching model. We can approximate a continuous-time non-linear dynamical system with latent dynamics $\dot{\boldsymbol{x}} = \boldsymbol{f}(\boldsymbol{x})$ around reference points $\{\tilde{\boldsymbol{x}}_k\}_{k=1}^K$ using a first-order Taylor expansion, $\boldsymbol{f}(\boldsymbol{x}) \approx \tilde{\boldsymbol{f}}(\boldsymbol{x}) = \boldsymbol{f}(\tilde{\boldsymbol{x}}_k) + \boldsymbol{J_f}(\tilde{\boldsymbol{x}}_k)(\boldsymbol{x} - \tilde{\boldsymbol{x}}_k)$, where we denote the Jacobian matrix of $\boldsymbol{f}$ with $\boldsymbol{J_f}$. We evaluate the equation at each point $t$ using the best reference point $\tilde{\boldsymbol{x}}_k$. We obtain system matrices $\boldsymbol{A}_k = \boldsymbol{J_f}(\tilde{\boldsymbol{x}}_k)$ and bias term $\boldsymbol{b}_k = \boldsymbol{f}(\tilde{\boldsymbol{x}}_k) - \boldsymbol{J_f}(\tilde{\boldsymbol{x}}_k)\tilde{\boldsymbol{x}}_k$ which can be modeled with the $\nabla$-SLDS model $\hat{\boldsymbol{f}}(\boldsymbol{x}_t; k_t)$:

$$\boldsymbol{x}_{t+1} = (\boldsymbol{A}_{k_t}\boldsymbol{x}_t + \boldsymbol{b}_{k_t}) + \boldsymbol{\varepsilon}_t =: \hat{\boldsymbol{f}}(\boldsymbol{x}_t; k_t) + \boldsymbol{\varepsilon}_t. \tag{8}$$

While a theoretical guarantee for this general case is beyond the scope of this work, we give an empirical evaluation on Lorenz attractor dynamics below. Note, as the number of "basis points" of $\nabla$-SLDS approaches the number of time steps, we could trivially approach perfect estimation capability of the latents as we store the exact value of $\boldsymbol{f}$ at every point. However, this comes at the expense of having less points to estimate each individual dynamics matrix. Empirically, we used 100–200 matrices for datasets of 1M samples.

## 5 EXPERIMENTS

To verify our theory, we implement a benchmark dataset for studying the effects of various model choices. We generate time-series with 1M samples, either as a single sequence or across multiple trials. Our experiments rigorously evaluate different variants of contrastive learning algorithms.

**Data generation.** Data is generated by simulating latent variables $\boldsymbol{x}$ that evolve according to a dynamical system (Eq. 5). These latent variables are then passed through a nonlinear mixing function $\boldsymbol{g}$ to produce the observable data $\boldsymbol{y}$. The mixing function $\boldsymbol{g}$ consists of a nonlinear injective component which is parameterized by a randomly initialized 4-layer MLP (Hyvarinen & Morioka, 2016), and a linear map to a 50-dimensional space. The final mixing function is defined as their composition. We ensure the injectivity of the resulting function by monitoring the condition number of each matrix layer, following previous work (Hyvarinen & Morioka, 2016; Zimmermann et al., 2021).

**LDS.** We simulate 1M datapoints in 3D space following $\boldsymbol{f}(\boldsymbol{x}_t) = \boldsymbol{A}\boldsymbol{x}_t$ with system noise standard deviation $\sigma_\epsilon = 0.01$ and choose $\boldsymbol{A}$ to be an orthogonal matrix to ensure stable dynamics with all eigenvalues equal to 1. We do so by taking the product of multiple rotation matrices, one for each possible plane to rotate around with rotation angles being randomly chosen to be -5° or 5°.

**SLDS.** We simulate switching linear dynamical systems with $\boldsymbol{f}(\boldsymbol{x}_t; k_t) = \boldsymbol{A}_{k_t}\boldsymbol{x}_t$ and system noise standard deviation $\sigma_\epsilon = 0.0001$. We choose $\boldsymbol{A}_k$ to be an orthogonal matrix ensuring that all eigenvalues are 1, which guarantees system stability. Specifically, we set $\boldsymbol{A}_k$ to be a rotation matrix with varying rotation angles (5°, 10°, 20°). The latent dimensionality is 6. The number of samples is 1M. We use 1000 trials, and each trial consists of 1000 samples. We use $k = 0, 1, \ldots, K$ distinct modes following a mode sequence $i_t$. The mode sequence $i_t$ follows a Markov chain with a symmetric transition matrix and uniform prior: $i_0 \sim \text{Cat}(\pi)$, where $\pi_j = \frac{1}{K}$ for all $j$. At each time step, $i_{t+1} \sim \text{Cat}(\Pi_{i_t})$, where $\Pi$ is a transition matrix with uniform off-diagonal probabilities set to $10^{-4}$. Example data is visualized in Figure 4 and Appendix E.

**Non-linear dynamics.** We simulate 1M points of a Lorenz system, with equations

$$\boldsymbol{f}(\boldsymbol{x}_t) = \boldsymbol{x}_t + dt[\sigma(x_{2,t} - x_{1,t}), x_{1,t}((\rho - x_{3,t}) - x_{2,t}), (x_{1,t}x_{2,t} - \beta x_{3,t})]^\top \tag{9}$$

with varying $dt$, parameters $\sigma = 10$, $\beta = \frac{8}{3}$, $\rho = 28$ and system noise standard deviation $\sigma_\epsilon = 0.001$. The observable data, $\boldsymbol{y}$. We then apply our non-linear mixing function as for other datasets.

**Model estimation.** For the feature encoder $\boldsymbol{h}$, baseline and our model use an MLP with three layers followed by GELU activations (Hendrycks & Gimpel, 2016). Model capacity scales with the embedding dimensionality $d$. The last hidden layer has $10d$ units and all previous layers have $30d$ units. For the SLDS and LDS datasets, we train on batches with 2048 samples each (reference

Table 1: Overview about identifiability of latent dynamics for different modeling choices: We show different *data* generating processes characterized by the form of the ground truth dynamics $\boldsymbol{f}$, the distribution $p(\boldsymbol{\varepsilon})$ and different *model* choices for the estimated dynamics $\hat{\boldsymbol{f}}$. We compare identity dynamics, linear dynamics (LDS), switching linear dynamics (SLDS), and Lorenz attractor dynamics (Lorenz), and optionally initialize the dynamics model with the ground-truth dynamics (GT). For every combination we indicate whether we can provide theoretical identifiability guarantees ("theory") and compare this to empirical identifiability measures ($R^2$, LDS, dynR$^2$). Mean $\pm$ std. are across 3 datasets (5 for Lorenz) and 3 experiment repeats.

| Data | | Model | Results | | |
| $\boldsymbol{f}$ | $p(\boldsymbol{\varepsilon})$ | $\hat{\boldsymbol{f}}$ | identifiable | $\%R^2 \uparrow$ | LDS $[\times 10^{-2}] \downarrow$ |
|---|---|---|---|---|---|
| identity | Normal | identity | ✓ | $99.56 \pm 0.21$ | $0.00 \pm 0.00$ |
| identity | Normal | LDS | ✓ | $99.31 \pm 0.43$ | $0.04 \pm 0.01$ |
| LDS (low $\Delta t$) | Normal (large $\sigma$) | identity | – | $89.22 \pm 4.47$ | $8.53 \pm 0.05$ |
| LDS | Normal | identity | ✗ | $73.56 \pm 24.45$ | $21.24 \pm 0.31$ |
| LDS | Normal | LDS | ✓ | $99.03 \pm 0.41$ | $0.77 \pm 1.07$ |
| LDS | Normal | GT | ✓ | $99.46 \pm 0.39$ | $0.44 \pm 0.43$ |
| | | | | $\%R^2 \uparrow$ | $\%\mathrm{dyn}R^2 \uparrow$ |
| SLDS | Normal | identity | ✗ | $76.80 \pm 7.40$ | $85.47 \pm 8.07$ |
| SLDS | Normal | $\nabla$-SLDS | (✓)[1] | $99.52 \pm 0.05$ | $99.93 \pm 0.01$ |
| SLDS | Normal | GT | (✓)[1] | $99.20 \pm 0.10$ | $99.97 \pm 0.00$ |
| Lorenz (small $\Delta t$) | Normal (large $\sigma$) | identity | – | $99.74 \pm 0.36$ | $99.94 \pm 0.07$ |
| Lorenz (small $\Delta t$) | Normal (large $\sigma$) | LDS | – | $98.31 \pm 2.55$ | $97.21 \pm 5.90$ |
| Lorenz (small $\Delta t$) | Normal (large $\sigma$) | $\nabla$-SLDS | – | $94.14 \pm 4.34$ | $94.20 \pm 6.57$ |
| Lorenz | Normal | identity | ✗ | $40.99 \pm 8.58$ | $27.02 \pm 8.72$ |
| Lorenz | Normal | LDS | ✗ | $81.20 \pm 16.93$ | $80.30 \pm 14.13$ |
| Lorenz | Normal | $\nabla$-SLDS | (✓)[2] | $94.08 \pm 2.75$ | $93.91 \pm 5.32$ |

and positive). We use $2^{16} = 65536$ negative samples for SLDS and $20k$ negative samples for LDS data. For the Lorenz data, we use a batch size of 1024 and 20k negative samples. We use the Adam optimizer (Kingma, 2014) with learning rates $3 \times 10^{-4}$ for LDS data, $10^{-3}$ for SLDS data, and $10^{-4}$ for Lorenz system data. For the SLDS data, we use a different learning rate of $10^{-2}$ for the parameters of the dynamics model. We train for 50k steps on SLDS data and for 30k steps for LDS and Lorenz system data. Our baseline model is standard self-supervised contrastive learning with the InfoNCE loss, which is similar to the CEBRA-time model (with symmetric encoders, i.e., without a dynamics model; cf. Schneider et al., 2023). For DCL, we add an LDS or $\nabla$-SLDS dynamics model for fitting. For our baseline, we post-hoc fit the corresponding model on the recovered latents minimizing the predictive mean squared error via gradient descent.

**Evaluation metrics.** Our metrics are informed by the result in Theorem 1 and measure empirical identifiability up to affine transformation of the latent space and its underlying linear or non-linear dynamics. All metrics are estimated on the dataset the model is fit on. See Appendix F for additional discussion on estimating metrics on independently sampled dynamics.

To account for the affine indeterminacy, we estimate $\boldsymbol{L}, \boldsymbol{b}$ for $\hat{\boldsymbol{x}} = \boldsymbol{L}\boldsymbol{x} + \boldsymbol{b}$ which allows us to map ground truth latents $\boldsymbol{x}$ to recovered latents $\hat{\boldsymbol{x}}$ (cf. Theorem 1a). In cases where the inverse transform $\boldsymbol{x} = \boldsymbol{L}^{-1}(\hat{\boldsymbol{x}} - \boldsymbol{b})$ is required, we can either compute $\boldsymbol{L}^{-1}$ directly, or for the purpose of numerical stability estimate it from data, which we denote as $\boldsymbol{L}'$. The values of $\boldsymbol{L}, \boldsymbol{b}$ and $\boldsymbol{L}', \boldsymbol{b}'$ are computed via a linear regression:

$$\min_{\boldsymbol{L},\boldsymbol{b}} \sum_{t=1}^{T} \|\hat{\boldsymbol{x}}_t - (\boldsymbol{L}\boldsymbol{x}_t + \boldsymbol{b})\|_2^2 \qquad \text{and} \qquad \min_{\boldsymbol{L}',\boldsymbol{b}'} \sum_{t=1}^{T} \|\boldsymbol{x}_t - (\boldsymbol{L}'\hat{\boldsymbol{x}}_t + \boldsymbol{b}')\|_2^2. \qquad (10)$$

To evaluate the identifiability of the representation, we measure the $R^2$ between the true latents $\boldsymbol{x}_t$ and the optimally aligned recovered latents $\boldsymbol{L}'\hat{\boldsymbol{x}}_t + \boldsymbol{b}'$ across time steps $t = 1 \dots T$ in the time-series.

---

[1] Not explicitly shown, but the argument in Corollary 2 applies to each piecewise linear section of the SLDS.
[2] $\nabla$-SLDS is only an approximation of the functional form of the underlying system.

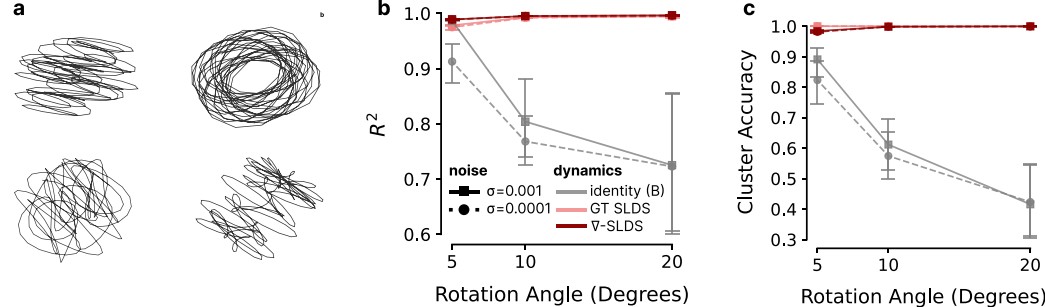

Figure 4: Switching linear dynamics: (a) example ground-truth dynamics in latent space for four matrices $\boldsymbol{A}_k$. (b) $R^2$ metric for different noise levels as we increase the angles used for data generation. We compare a baseline (no dynamics) to $\nabla$-SLDS and a model fitted with ground-truth dynamics. (c) cluster accuracies for models shown in (b).

We also propose two metrics as direct measures of identifiability for the recovered dynamics $\hat{\boldsymbol{f}}$. For linear dynamics models, we introduce the LDS error. It denotes the norm of the difference between the true dynamics matrix $\boldsymbol{A}$ and the estimated dynamics matrix $\hat{\boldsymbol{A}}$ by accounting for the linear transformation between the true and recovered latent spaces. The LDS error (related to the metric for Dynamical Similarity Analysis; Ostrow et al., 2023) is computed as (cf. Corollary 2):

$$\text{LDS}(\boldsymbol{A}, \hat{\boldsymbol{A}}) = \|\boldsymbol{A} - \boldsymbol{L}^{-1}\hat{\boldsymbol{A}}\boldsymbol{L}\|_F. \tag{11}$$

As a second, more general identifiability metric for the recovered dynamics $\hat{\boldsymbol{f}}$, we introduce $\text{dyn}R^2$, which builds on Theorem 1b to evaluate the identifiability of non-linear dynamics. This metric computes the $R^2$ between the predicted dynamics $\hat{\boldsymbol{f}}$ and the true dynamics $\boldsymbol{f}$, corrected for the linear transformation between the two latent spaces. Specifically, motivated by Theorem 1(b), we compute

$$\text{dyn}R^2(\boldsymbol{f}, \hat{\boldsymbol{f}}) = \text{r2\_score}(\hat{\boldsymbol{f}}(\hat{\boldsymbol{x}}), \boldsymbol{L}\boldsymbol{f}(\boldsymbol{L}'\hat{\boldsymbol{x}} + \boldsymbol{b}') + \boldsymbol{b}) \tag{12}$$

along all time steps. Additional variants of the $\text{dyn}R^2$ metric are discussed in Appendix G.

Finally, when evaluating switching linear dynamics, we compute the accuracy for assigning the correct mode at any point in time. To compute the cluster accuracy in the case of SLDS ground truth dynamics, we leverage the Hungarian algorithm to match the estimated latent variables modeling mode switches to the ground truth modes, and then proceed to compute the accuracy.

**Implementation.** Experiments were carried out on a compute cluster with A100 cards. On each card, we ran $\sim$3 experiments simultaneously. Depending on the exact configuration, training time varied from 5–20min per model. The combined experiments ran for this paper comprised about 120 days of A100 compute time and we provide a breakdown in Appendix K. We will open source our benchmark suite for identifiable dynamics learning upon publication of the paper.

## 6 RESULTS

### 6.1 VERIFICATION OF THE THEORY FOR LINEAR DYNAMICS

**Suitable dynamics models enable identification of latents and dynamics.** For all considered classes of models, we show in Table 1 that DCL with a suitable dynamics model effectively identifies the correct dynamics. For linear dynamics (LDS), DCL reaches an $R^2$ of 99.0%, close to the oracle performance (99.5%). Most importantly, the average LDS error of our method ($7.7 \times 10^{-3}$) is very close to the oracle ($4.4 \times 10^{-3}$), in contrast to the baseline model ($2.1 \times 10^{-1}$) which has a substantially larger LDS error. In the case of switching linear dynamics (SLDS), DCL also shows strong performance, both in terms of latent $R^2$ (99.5%) and dynamics $R^2$ (99.9%) outperforming the respective baselines (76.8% $R^2$ and 85.5% dynamics $R^2$). For non-linear dynamics, the baseline model fails entirely (41.0%/27.0%), while $\nabla$-SLDS dynamics can be fitted with 94.1% $R^2$ for latents and 93.9% dynamics $R^2$. We also clearly see the strength of our piecewise-linear approximation, as the LDS dynamics models only reaches 81.2% latent identifiability and 80.3% dynamics $R^2$.

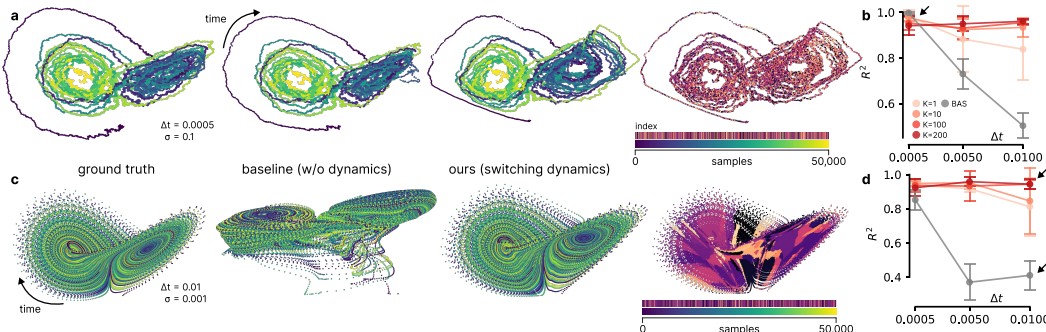

Figure 5: Contrastive learning of 3D non-linear dynamics following a Lorenz attractor model. (a), left to right: ground truth dynamics for 10k samples with $dt = 0.0005$ and $\sigma = 0.1$, estimation results for baseline (identity dynamics), DCL with $\nabla$-SLDS, estimated mode sequence. (b), empirical identifiability ($R^2$) between baseline (BAS) and $\nabla$-SLDS for varying numbers of discrete states $K$. (c, d), same layout but for $dt = 0.01$ and $\sigma = 0.001$.

**Learning noisy dynamics does not require a dynamics model.** If the variance of the distribution for $\varepsilon_t$ dominates the changes actually introduced by the dynamics, we find that the baseline model is also able to identify the latent space underlying the system. Intuitively, the change introduced by the dynamical system is then negligible compared to the noise. In Table 1 ("large $\sigma$"), we show that recovery is possible for cases with small angles, both in the linear and non-linear case. While in some cases, this learning setup might be applicable in practice, it seems generally unrealistic to be able to perturb the system beyond the actual dynamics. As we scale the dynamics to larger values (Figure 4, panel b and c), the estimation scheme breaks again. However, this property offers an explanation for the success of existing contrastive estimation algorithms like CEBRA-time (Schneider et al., 2023) which successfully estimate dynamics in absence of a dynamics model.

**Symmetric encoders cannot identify non-trivial dynamics.** In the more general case where the dynamics dominates the system behavior, the baseline cannot identify linear dynamics (or more complicated systems). In the general LDS and SLDS cases, the baseline fails to identify the ground truth dynamics (Table 1) as predicted by Corollary 1 (rows marked with ✗). For identity dynamics, the baseline is able to identify the latents ($R^2$=99.56%) but breaks as soon as linear dynamics are introduced ($R^2$=73.56%).

## 6.2 APPROXIMATION OF NON-LINEAR DYNAMICS

Next, we study in more details how the DCL can identify piecewise linear or non-linear latent dynamics using the $\nabla$-SLDS dynamics model.

**Identification of switching dynamics.** Switching dynamics are depicted in Fig. 4a for four different modes of the 10 degrees dataset. DCL obtains high $R^2$ for various choices of dynamics (Fig. 4b) and additionally identifies the correct mode sequence (Fig. 4c) for all noise levels and variants of the underlying dynamics. As we increase the rotation angle used to generate the matrices, the gap between baseline and our model increases substantially.

**Non-linear dynamics.** Figure 5 depicts the Lorenz system as an example of a non-linear dynamical system for different choices of algorithms. The ground truth dynamics vary in the ratio between $dt/\sigma$ and we show the full range in panels b/c. When the noise dominates the dynamics (panel a), the baseline is able to estimate also the nonlinear dynamics accurately, with 99.7%. However, as we move to lower noise cases (panel b), performance reduces to 41.0%. Our switching dynamics model is able to estimate the system with high $R^2$ in both cases (94.14% and 94.08%). However, note that in this non-linear case, we are primarily succeeding at estimating the latent space, the estimated dynamics model did not meaningfully outperform an identity model (Appendix G).

**Extensions to other distributions $p_\varepsilon$.** While Euclidean geometry is most relevant for dynamical systems in practice, and hence the focus of our theoretical and empirical investigation, contrastive learning commonly operates on the hypersphere in other contexts. We provide additional results for the case of a von Mises-Fisher (vMF) distribution for $p_\varepsilon$ and dot-product similarity for $\phi$ in Appendix D.

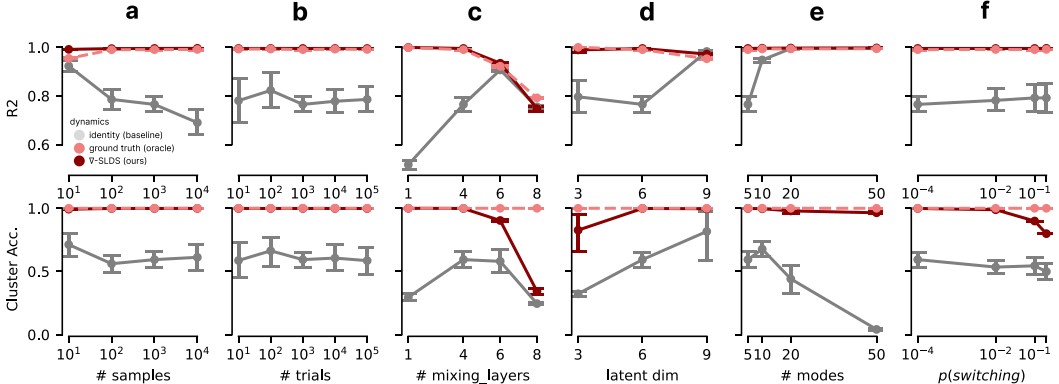

Figure 6: Variations and ablations for the SLDS. We compare the $\nabla$-SLDS model to the ground-truth switching dynamics (oracle) and a standard CL model without dynamics (baseline). All variations are with respect to the setting with 1M time steps (1k trials $\times$ 1k samples), $L = 4$ mixing layers, $d = 6$ latent dimensionality, 5 modes, and $p = 0.0001$ switching probability. We study the impact of the dataset size in terms of (a) samples per trial, (b) the number of trials, the impact of nonlinearity of the observations in terms of (c) number of mixing layers, the impact of complexity of the latent dynamics in terms of (d) latent dimensionality, (e) number of modes to switch in between and (f) the switching frequency paramameterized via the switching probability.

## 6.3 ABLATION STUDIES

For practitioners leveraging contrastive learning for statistical analysis, it is important to know the trade-offs in empirical performance in relation to various parameters. In real-world experiments, the most important factors are the size of the dataset, the trial-structure of the dataset, the latent dimensionality we can expect to recover, and the degree of non-linearity between latents and observables. We consider these factors of influence: As a reference, we use the SLDS system with a 6D latent space, 1M samples (1k trials $\times$ 1k samples), $L = 4$ mixing layers, 10 degrees for the rotation matrices, 65,536 negative samples per batch; batch size 2,048, and learning rate $10^{-3}$.

**Impact of dataset size (Fig. 6a).** We keep the number of trials fixed to 1k. As we vary the sample size per trial, $R^2$ degrades for smaller dataset, and for the given setting we need at least 100 points per trial to attain identifiability empirically. We outperform the baseline model in all cases.

**Impact of trials (Fig. 6b).** We next simulate a fixed number of 1M datapoints, which we split into trials of varying length. We consider 1k, 10k, 100k, and 1M as trial lengths. Performance is stable for the different settings, even for cases with small trial length (and less observed switching points). DCL consistently outperforms the baseline algorithm and attains stable performance close to the theoretical maximum given by the ground-truth dynamics.

**Impact of non-linear mixing (Fig. 6c).** All main experiments have been conducted with $L = 4$ mixing layers in the mixing function $\boldsymbol{g}$. Performance of DCL stays at the theoretical maximum as we increase the number of mixing layers. As we move beyond four layers, both oracle performance in $R^2$ and our model declines, hinting that either (1) more data or (2) a larger model is required to recover the dynamics successfully in these cases.

**Impact of dimensionality (Fig. 6d).** Increasing latent dimensionality does not meaningfully impact performance of our model. We found that for higher dimensions, it is crucial to use a large number of negative examples (65k) for successful training.

**Number of modes for switching linear dynamics fitting (Fig. 6e).** Increasing the number of modes in the dataset leads to more successful fitting of the $R^2$ for the baseline model, but to a decline in accuracy. This might be due to the increased variance: While this

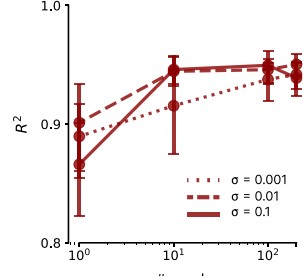

Figure 7: Impact of modes for non-linear dynamics in the Lorenz system for different system noise levels $\sigma$, averaged over all $dt$.

helps the model to identify the latent space (dynamics appear more like noise), it still fails to identify the underlying dynamics model, unlike DCL which attains high $R^2$ and cluster accuracy throughout.

**Robustness to changes in switching probability (Fig. 6f).** Finally, we vary the switching probability. Higher switching probability causes shorter modes, which are harder to fit by the $\nabla$-SLDS dynamics model. Our model obtains high empirical identifiability throughout the experiment, but the accuracy metric begins to decline when $p = 0.1$ and $p = 0.2$.

**Number of modes for non-linear dynamics fitting (Fig. 7).** We study the effect of increasing the number of matrices in the parameter bank $\mathbf{W}$ in the $\nabla$-SLDS model. The figure depicts the impact of increasing the number of modes for DCL on the non-linear Lorenz dataset. We observe that increasing modes to 200 improves performance, but eventually converges to a stable maximum for all noise levels.

## 7 DISCUSSION

The DCL framework is versatile and allows to study the performance of contrastive learning in conjunction with different dynamics models. By exploring various special cases (identity, linear, switching linear), our study categorizes different forms of contrastive learning and makes predictions about their behavior in practice. In comparison to contrastive predictive coding (CPC; Oord et al., 2018) or wav2vec (Schneider et al., 2019), DCL generalizes the concept of training contrastive learning models with (explicit) dynamics models. CPC uses an RNN encoder followed by linear projection, while wav2vec leverages CNNs dynamics models and affine projections. Theorem 1 applies to both these models, and offers an explanation for their successful empirical performance.

Nonlinear ICA methods, such as TCL (Hyvarinen & Morioka, 2016) and PCL (Hyvarinen & Morioka, 2017) provide identifiability of the latent variables leveraging temporal structure of the data. Compared to DCL, they do not explicitly model dynamics and assume either stationarity or non-stationarity of the time series (Hyvärinen et al., 2023), whereas DCL assumes bijective latent dynamics, and focuses on explicit dynamics modeling beyond solving the demixing problem.

For applications in scientific data analysis, CEBRA (Schneider et al., 2023) uses supervised or self-supervised contrastive learning, either with symmetric encoders or asymmetric encoder functions. While our results show that such an algorithm is able to identify dynamics for a sufficient amount of system noise, adding dynamics models is required as the system dynamics dominate. Hence, the DCL approach with LDS or $\nabla$-SLDS dynamics generalises the self-supervised mode of CEBRA and makes it applicable for a broader class of problems.

Finally, there is a connection to the joint embedding predictive architecture (JEPA; LeCun, 2022; Assran et al., 2023). The architecture setup of DCL can be regarded as a special case of JEPA, but with symmetric encoders to leverage distillation of the system dynamics into the predictor (the dynamics model). In contrast to JEPA, the use of symmetric encoders requires a contrastive loss for avoiding collapse and, more importantly, serves as the foundation for our theoretical result.

A limitation of the present study is its main focus on simulated data which clearly corroborates our theory but does not yet demonstrate real-world applicability. However, our simulated data bears the signatures of real-world datasets (multi-trial structures, varying degrees of dimensionality, number of modes, and different forms of dynamics). A challenge is the availability of real-world benchmark datasets for dynamics identification. We believe that rigorous evaluation of different estimation methods on such datasets will continue to show the promise of contrastive learning for dynamics identification. Integrating recent benchmarks like DynaDojo (Bhamidipaty et al., 2023) or datasets from Chen et al. (2021) with realistic mixing functions ($\boldsymbol{g}$) offers a promising direction for evaluating latent dynamics models. As a demonstration of real-world applicability, we benchmarked DCL on a neural recordings dataset in Appendix J.

## 8 CONCLUSION

We proposed a first identifiable, end-to-end, non-generative inference algorithm for latent switching dynamics along with an empirically successful parameterization of non-linear dynamics. Our results point towards the empirical effectiveness of contrastive learning across time-series, and back these empirical successes by theory. We show empirical identifiability with limited data for linear, switching linear and non-linear dynamics. Our results add to the understanding of SSL's empirical success, will guide the design of future contrastive learning algorithms and most importantly, make SSL amenable for computational statistics and data analysis.

REPRODUCIBILITY STATEMENT

**Code.** Code is available at `https://github.com/dynamical-inference/dcl` under an Apache 2.0 license. Experimental and implementation details for the main text are given in section 5 and for each experiment of the Appendix within the respective chapter.

**Theory.** Our theoretical claims are backed by a complete proof attached in Appendix A. Assumptions are outlined in the main text (Section 3) and again in more detail in Appendix A.

**Datasets.** We evaluate our experiments on a variety of synthetic datasets. The datasets comprise different dynamical systems, from linear to nonlinear.

**Compute.** Moderate compute resources are required to reproduce this paper. As stated in section 5, we used around 120 days of GPU compute on a A100 to produce the results presented in the paper. We provide a more detailed breakdown in Appendix K.

AUTHOR CONTRIBUTIONS

RGL and TS: Methodology, Software, Investigation, Writing–Editing. StS: Conceptualization, Methodology, Formal Analysis, Writing–Original Draft and Writing–Editing.

ACKNOWLEDGMENTS

We thank Luisa Eck and Stephen Jiang for discussions on the theory, and Lilly May for input on paper figures. We thank the five anonymous reviewers at ICLR for their valuable and constructive comments on our manuscript. This work was supported by the Helmholtz Association's Initiative and Networking Fund on the HAICORE@KIT and HAICORE@FZJ partitions.

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

# Supplementary Material

## A  Proof of the main result

We re-state Theorem 1 from the main paper, and provide a full proof below:

**Theorem 1** (Contrastive estimation of non-linear dynamics). *Assume that*

- *(A1) A time-series dataset $\{\boldsymbol{y}_t\}_{t=1}^T$ is generated according to the ground-truth dynamical system in Eq. 5 with a bijective dynamics model $\boldsymbol{f}$ and an injective mixing function $\boldsymbol{g}$.*
- *(A2) The system noise follows an iid normal distribution, $p(\boldsymbol{\varepsilon}_t) = \mathcal{N}(\boldsymbol{\varepsilon}_t|0, \boldsymbol{\Sigma_\varepsilon})$.*
- *(A3) The model $\psi$ is composed of an encoder $\boldsymbol{h}$, a dynamics model $\hat{\boldsymbol{f}}$, a correction term $\alpha$, and the similarity metric $\phi(\boldsymbol{u}, \boldsymbol{v}) = -\|\boldsymbol{u} - \boldsymbol{v}\|^2$ and attains the global minimizer of Eq. 3.*

*Then, in the limit of $T \to \infty$ for any point $\boldsymbol{x}$ in the support of the data marginal distribution:*

- *(a) The composition of mixing and de-mixing $\boldsymbol{h}(\boldsymbol{g}(\boldsymbol{x})) = \boldsymbol{L}\boldsymbol{x} + \boldsymbol{b}$ is a bijective affine transform, and $\boldsymbol{L} = \boldsymbol{Q}\boldsymbol{\Sigma}_\epsilon^{-1/2}$ with unknown orthogonal transform $\boldsymbol{Q} \in \mathbb{R}^{d \times d}$ and offset $\boldsymbol{b} \in \mathbb{R}^d$.*
- *(b) The estimated dynamics $\hat{\boldsymbol{f}}$ are bijective and identify the true dynamics $\boldsymbol{f}$ up to the relation $\hat{\boldsymbol{f}}(\boldsymbol{x}) = \boldsymbol{L}\boldsymbol{f}(\boldsymbol{L}^{-1}(\boldsymbol{x} - \boldsymbol{b})) + \boldsymbol{b}$.*

*Proof.* Our proof proceeds in three steps: First, we leverage existing theory (Wang & Isola, 2020; Zimmermann et al., 2021) to arrive at the minimizer of the contrastive loss, and relate the limited sample loss function to the asymptotic case. Second, we derive the statement about achieving successful demixing in Theorem 1(a). Finally, we derive the statement in Theorem 1(a) about structural identifiability of the dynamics model.

**Step 1: Minimizer of the InfoNCE loss.** By the assumption $\boldsymbol{\varepsilon}_t$ is normally distributed, we obtain the positive sample conditional distribution

$$p(\boldsymbol{x}_{t+1}|\boldsymbol{x}_t) = \mathcal{N}(\boldsymbol{x}_{t+1}|\boldsymbol{f}(\boldsymbol{x}_t), \boldsymbol{\Sigma_\varepsilon}). \tag{13}$$

The negative sample distribution $q(\boldsymbol{x}_t)$ is obtained by sampling $t$ uniformly from all time steps and can hence be written as a Gaussian mixture along the dynamics imposed by $\boldsymbol{f}$,

$$q(\boldsymbol{x}) = \frac{1}{T} \sum_{t=1}^T p_{\boldsymbol{\varepsilon}}(\boldsymbol{x} - \boldsymbol{x}_t) \tag{14}$$

$$= \frac{1}{T} \sum_{t=1}^T \mathcal{N}(\boldsymbol{x} - \boldsymbol{x}_t | \boldsymbol{f}(\boldsymbol{x}_{t-1}), \boldsymbol{\Sigma_\varepsilon}). \tag{15}$$

We use these definitions of $p$ and $q$ to study the asymptotic case of our loss function. For $T \to \infty$, due to Wang & Isola (2020) we can rewrite the limit of our loss function (Eq. 3) as

$$\mathcal{L}[\psi] = \lim_{T \to \infty} \mathbb{E}_{t,N}[-\log p_\psi(\boldsymbol{y}_{t+1}|\boldsymbol{y}_t, N)] - \log T$$

$$= \int q(\boldsymbol{y}) \left[ -\int p(\boldsymbol{y}'|\boldsymbol{y})\psi(\boldsymbol{y}, \boldsymbol{y}')d\boldsymbol{y}' + \log \int q(\boldsymbol{y}') \exp[\psi(\boldsymbol{y}, \boldsymbol{y}')]d\boldsymbol{y}' \right]. \tag{16}$$

It was shown (Proposition 1, Schneider et al., 2023) that this loss function is convex in $\psi$ with the unique minimizer

$$\psi(\boldsymbol{g}(\boldsymbol{x}), \boldsymbol{g}(\boldsymbol{x}')) = \log \frac{p(\boldsymbol{x}'|\boldsymbol{x})}{q(\boldsymbol{x}')} + c(\boldsymbol{x}), \tag{17}$$

where $c : \mathbb{R}^d \mapsto \mathbb{R}$ is an arbitrary scalar-valued function. Note that we also expressed $\boldsymbol{y} = \boldsymbol{g}(\boldsymbol{x}), \boldsymbol{y}' = \boldsymbol{g}(\boldsymbol{x}')$ to continue the proof in terms of the relation between original and estimated latents. We insert the definition of the model on the left hand side. Let us also denote $\boldsymbol{h} \circ \boldsymbol{g} =: \boldsymbol{r}$, and the definition of the ground-truth generating process on the right hand side to obtain

$$\phi(\hat{\boldsymbol{f}}(\boldsymbol{r}(\boldsymbol{x})), \boldsymbol{r}(\boldsymbol{x}')) - \alpha(\boldsymbol{x}') = \log p(\boldsymbol{x}'|\boldsymbol{f}(\boldsymbol{x})) - \log q(\boldsymbol{x}') + c(\boldsymbol{x}). \tag{18}$$

Inserting the potential[2] as $\alpha(\boldsymbol{x}) = \log q(\boldsymbol{x})$ simplifies the equation to

$$\phi(\hat{\boldsymbol{f}}(\boldsymbol{r}(\boldsymbol{x})), \boldsymbol{r}(\boldsymbol{x}')) = \log p(\boldsymbol{x}'|\boldsymbol{f}(\boldsymbol{x})) + c(\boldsymbol{x}). \tag{19}$$

From here onwards, we will use that $\phi$ is the negative squared Euclidean norm (A.3) and correspondingly, the positive conditional is a normal distribution with concentration $\boldsymbol{\Lambda} = \boldsymbol{\Sigma}_u^{-1}$ (A.2),

$$-\|\hat{\boldsymbol{f}}(\boldsymbol{r}(\boldsymbol{x})) - \boldsymbol{r}(\boldsymbol{x}')\|_2^2 = -(\boldsymbol{f}(\boldsymbol{x}) - \boldsymbol{x}')^\top \boldsymbol{\Lambda}(\boldsymbol{f}(\boldsymbol{x}) - \boldsymbol{x}') + c'(\boldsymbol{x}) \tag{20}$$

where we pulled the normalization constant of $p$ into the function $c'$ for brevity.

**Step 2: Properties of the feature encoder.** Starting from the last equation, we compute the derivative with respect to $\boldsymbol{x}$ and $\boldsymbol{x}'$ on both sides and obtain

$$\boldsymbol{J}_r^\top(\boldsymbol{x}')\boldsymbol{J}_{\hat{f}}(\boldsymbol{r}(\boldsymbol{x}))\boldsymbol{J}_r(\boldsymbol{x}) = \boldsymbol{\Lambda}\boldsymbol{J}_f(\boldsymbol{x}). \tag{21}$$

Because this equation holds for any $\boldsymbol{x}' \in \operatorname{supp} q$ independently of $\boldsymbol{x}$, we can conclude that the Jacobian matrix of $\boldsymbol{r}$ needs to be constant. From there it follows that $\boldsymbol{r}$ is affine. Let us write $\boldsymbol{r}(\boldsymbol{x}) = \boldsymbol{L}\boldsymbol{x} + \boldsymbol{b}$. Then, the Jacobian matrix is $\boldsymbol{J}_r = \boldsymbol{L}$. Inserting this yields

$$\boldsymbol{L}^\top \boldsymbol{J}_{\hat{f}}(\boldsymbol{L}\boldsymbol{x} + \boldsymbol{b})\boldsymbol{L} = \boldsymbol{\Lambda}\boldsymbol{J}_f(\boldsymbol{x}). \tag{22}$$

We next establish that $\boldsymbol{L}$ has full rank: because the dynamics function $\boldsymbol{f}$ is bijective by assumption, $\boldsymbol{J}_f(\boldsymbol{x})$ has full rank $d$. $\boldsymbol{\Lambda}$ has full rank by assumption about the distribution $p_\varepsilon(\boldsymbol{\varepsilon})$. All matrices on the LHS are square and need to have full rank as well for any point $\boldsymbol{x}$. Hence, we can conclude that $\boldsymbol{L}$ has full rank, and likewise $\boldsymbol{J}_{\hat{f}}$. From there, we conclude that $\boldsymbol{r}$ and $\hat{\boldsymbol{f}}$ are bijective.

Next, we derive additional constraints on the matrix $\boldsymbol{L}$. We insert the solution for $\boldsymbol{r}$ obtained so far in Eq. 20,

$$-\|\hat{\boldsymbol{f}}(\boldsymbol{L}\boldsymbol{x} + \boldsymbol{b}) - \boldsymbol{L}\boldsymbol{x}' - \boldsymbol{b}\|^2 = -(\boldsymbol{f}(\boldsymbol{x}) - \boldsymbol{x}')^\top \boldsymbol{\Lambda}(\boldsymbol{f}(\boldsymbol{x}) - \boldsymbol{x}') + c'(\boldsymbol{x}) \tag{23}$$

and take the derivative twice with respect to $\boldsymbol{x}'$, to obtain

$$\boldsymbol{L}^\top \boldsymbol{L} = \boldsymbol{\Lambda} \quad \Leftrightarrow \quad \boldsymbol{L}^\top = \boldsymbol{\Lambda}\boldsymbol{L}^{-1}. \tag{24}$$

Without loss of generality, we introduce $\boldsymbol{Q} \in \mathbb{R}^{d \times d}$ to write $\boldsymbol{L}$ in terms of $\boldsymbol{\Lambda}$ as $\boldsymbol{L} = \boldsymbol{Q}\boldsymbol{\Lambda}^{1/2}$. Inserting into the previous equation lets us conclude

$$\boldsymbol{L}^\top \boldsymbol{L} = \boldsymbol{\Lambda}^{1/2}\boldsymbol{Q}^\top \boldsymbol{Q}\boldsymbol{\Lambda}^{1/2} = \boldsymbol{\Lambda} \tag{25}$$

$$\boldsymbol{Q}^\top \boldsymbol{Q} = \boldsymbol{\Lambda}^{-1/2}\boldsymbol{\Lambda}\boldsymbol{\Lambda}^{-1/2} = \boldsymbol{I}, \tag{26}$$

from which follows that $\boldsymbol{Q}$ is an orthogonal matrix. Hence, $\boldsymbol{L}$ is a composition of an orthogonal transform and $\boldsymbol{\Sigma}_\varepsilon^{-1/2}$, concluding the first part of the proof for statement (a).

**Step 3: Dynamics.** To derive part (b), we start at the condition Eq. 20 again to determine the value of $c'(\boldsymbol{x})$. We can consider two special cases:

$$\boldsymbol{f}(\boldsymbol{x}) = \boldsymbol{x}' : \quad c'(\boldsymbol{x}) = -\|\hat{\boldsymbol{f}}(\boldsymbol{r}(\boldsymbol{x})) - \boldsymbol{r}(\boldsymbol{x}')\|^2 \leq 0, \tag{27}$$

$$\hat{\boldsymbol{f}}(\boldsymbol{r}(\boldsymbol{x})) = \boldsymbol{r}(\boldsymbol{x}') : \quad c'(\boldsymbol{x}) = (\boldsymbol{f}(\boldsymbol{x}) - \boldsymbol{x}')^\top \boldsymbol{\Lambda}(\boldsymbol{f}(\boldsymbol{x}) - \boldsymbol{x}') \geq 0, \tag{28}$$

where we use that the concentration matrix $\boldsymbol{\Lambda}$ of a Normal distribution is positive semi-definite. When combining both conditions for points where $\boldsymbol{f}(\boldsymbol{x}) = \boldsymbol{x}'$ and $\hat{\boldsymbol{f}}(\boldsymbol{r}(\boldsymbol{x})) = \boldsymbol{r}(\boldsymbol{x}')$ the only admissible solution is $c'(\boldsymbol{x}) = 0$ for points with $\hat{\boldsymbol{f}}(\boldsymbol{r}(\boldsymbol{x})) = \boldsymbol{r}(\boldsymbol{f}(\boldsymbol{x}))$, i.e. $\hat{\boldsymbol{f}}(\boldsymbol{x}) = \boldsymbol{r}(\boldsymbol{f}(\boldsymbol{r}^{-1}(\boldsymbol{x})))$, hinting at the final solution. However, we have not shown yet that this solution is unique.

To show uniqueness, without loss of generality, we use the ansatz (with a residual $\boldsymbol{v}$)

$$\boldsymbol{f}(\boldsymbol{x}) = \boldsymbol{A}_1\hat{\boldsymbol{f}}(\boldsymbol{L}\boldsymbol{x} + \boldsymbol{b}) + \boldsymbol{d}_1 + \boldsymbol{v}(\boldsymbol{x}), \tag{29}$$

and computing the derivative with respect to $\boldsymbol{x}$ yields

$$\boldsymbol{J}_f(\boldsymbol{x}) = \boldsymbol{A}_1\boldsymbol{J}_{\hat{f}}(\boldsymbol{L}\boldsymbol{x} + \boldsymbol{b})\boldsymbol{L} + \boldsymbol{J}_v(\boldsymbol{x}). \tag{30}$$

---

[2]This is feasible in practice by parameterizing $\alpha(\boldsymbol{x})$ as a kernel density estimate, but empirically often not required. See Appendix B for additional technical details.

We insert this into Eq. 22 and obtain

$$\boldsymbol{L}^{\top}\boldsymbol{J}_{\hat{\boldsymbol{f}}}(\boldsymbol{L}\boldsymbol{x}+\boldsymbol{b})\boldsymbol{L} = \boldsymbol{\Lambda}\boldsymbol{A}_1\boldsymbol{J}_{\hat{\boldsymbol{f}}}(\boldsymbol{L}\boldsymbol{x}+\boldsymbol{b})\boldsymbol{L} + \boldsymbol{\Lambda}\boldsymbol{J}_{\boldsymbol{v}}(\boldsymbol{x}) \tag{31}$$

$$(\boldsymbol{L}^{\top} - \boldsymbol{\Lambda}\boldsymbol{A}_1)\boldsymbol{J}_{\hat{\boldsymbol{f}}}(\boldsymbol{L}\boldsymbol{x}+\boldsymbol{b})\boldsymbol{L} = \boldsymbol{\Lambda}\boldsymbol{J}_{\boldsymbol{v}}(\boldsymbol{x}) \tag{32}$$

$$(\boldsymbol{L}^{\top} - \boldsymbol{\Lambda}\boldsymbol{A}_1) = \boldsymbol{\Lambda}\boldsymbol{J}_{\boldsymbol{v}}(\boldsymbol{x})\boldsymbol{L}^{-1}\boldsymbol{J}_{\hat{\boldsymbol{f}}}^{-1}(\boldsymbol{L}\boldsymbol{x}+\boldsymbol{b}) \tag{33}$$

The left hand side is a constant, hence the same needs to hold true for the right hand side. Without loss of generality, let us introduce an arbitrary matrix $\boldsymbol{A}_2$ we set as this constant,

$$\boldsymbol{J}_{\boldsymbol{v}}(\boldsymbol{x})\boldsymbol{L}^{-1}\boldsymbol{J}_{\hat{\boldsymbol{f}}}^{-1}(\boldsymbol{L}\boldsymbol{x}+\boldsymbol{b}) = \boldsymbol{A}_2 \tag{34}$$

$$\boldsymbol{J}_{\boldsymbol{v}}(\boldsymbol{x}) = \boldsymbol{A}_2\boldsymbol{J}_{\hat{\boldsymbol{f}}}(\boldsymbol{L}\boldsymbol{x}+\boldsymbol{b})\boldsymbol{L} \tag{35}$$

which only admits the solution

$$\boldsymbol{v}(\boldsymbol{x}) = \boldsymbol{A}_2\hat{\boldsymbol{f}}(\boldsymbol{L}\boldsymbol{x}+\boldsymbol{b}) + \boldsymbol{d}_2, \tag{36}$$

where we introduced an additional integration constant $\boldsymbol{d}_2$. Inserting this into the ansatz in Eq. 29 gives

$$\boldsymbol{f}(\boldsymbol{x}) = \boldsymbol{A}_1\hat{\boldsymbol{f}}(\boldsymbol{L}\boldsymbol{x}+\boldsymbol{b}) + \boldsymbol{d}_1 + \boldsymbol{v}(\boldsymbol{x}), \tag{37}$$

$$\boldsymbol{f}(\boldsymbol{x}) = (\boldsymbol{A}_1 + \boldsymbol{A}_2)\hat{\boldsymbol{f}}(\boldsymbol{L}\boldsymbol{x}+\boldsymbol{b}) + (\boldsymbol{d}_1 + \boldsymbol{d}_2). \tag{38}$$

Using the shorthand $\boldsymbol{A} = \boldsymbol{A}_1 + \boldsymbol{A}_2$, $\boldsymbol{d} = \boldsymbol{d}_1 + \boldsymbol{d}_2$ we can repeat the steps in Eqs. 30–33 to arrive at the condition

$$(\boldsymbol{L}^{\top} - \boldsymbol{\Lambda}\boldsymbol{A})\boldsymbol{J}_{\hat{\boldsymbol{f}}}(\boldsymbol{L}\boldsymbol{x}+\boldsymbol{b})\boldsymbol{L} = 0. \tag{39}$$

Since all matrices have full rank, the only valid solution is $\boldsymbol{A} = \boldsymbol{\Lambda}^{-1}\boldsymbol{L}^{\top} = \boldsymbol{L}^{-1}$. Inserting back into the ansatz yields the refined solution

$$\boldsymbol{f}(\boldsymbol{x}) = \boldsymbol{L}^{-1}\hat{\boldsymbol{f}}(\boldsymbol{L}\boldsymbol{x}+\boldsymbol{b}) + \boldsymbol{d}, \tag{40}$$

and for brevity, we let $\boldsymbol{\xi} = \hat{\boldsymbol{f}}(\boldsymbol{r}(\boldsymbol{x})) = \hat{\boldsymbol{f}}(\boldsymbol{L}\boldsymbol{x}+\boldsymbol{b})$:

$$\boldsymbol{f}(\boldsymbol{x}) = \boldsymbol{L}^{-1}\boldsymbol{\xi} + \boldsymbol{d}. \tag{41}$$

We then insert the current solution into Eq. 20 and input $\boldsymbol{r}$ which gives

$$\|\boldsymbol{\xi} - \boldsymbol{L}\boldsymbol{x}' - \boldsymbol{b}\|^2 = (\boldsymbol{L}^{-1}\boldsymbol{\xi} + \boldsymbol{d} - \boldsymbol{x}')^{\top}\boldsymbol{\Lambda}(\boldsymbol{L}^{-1}\boldsymbol{\xi} + \boldsymbol{d} - \boldsymbol{x}') + c'(\boldsymbol{x}) \tag{42}$$

$$= (\boldsymbol{L}^{-1}\boldsymbol{\xi} + \boldsymbol{d} - \boldsymbol{x}')^{\top}\boldsymbol{L}^{\top}\boldsymbol{L}(\boldsymbol{L}^{-1}\boldsymbol{\xi} + \boldsymbol{d} - \boldsymbol{x}') + c'(\boldsymbol{x}) \tag{43}$$

$$= \|\boldsymbol{\xi} + \boldsymbol{L}\boldsymbol{d} - \boldsymbol{L}\boldsymbol{x}'\|^2 + c'(\boldsymbol{x}) \tag{44}$$

$$c'(\boldsymbol{x}) = \|\boldsymbol{\xi} - \boldsymbol{L}\boldsymbol{x}' - \boldsymbol{b}\|^2 - \|\boldsymbol{\xi} - \boldsymbol{L}\boldsymbol{x}' + \boldsymbol{L}\boldsymbol{d}\|^2 \tag{45}$$

Let us denote $\boldsymbol{v} = \boldsymbol{\xi} - \boldsymbol{L}\boldsymbol{x}'$ and note that $\boldsymbol{v}$ and $\boldsymbol{x}$ remain independent variables. We then get

$$c'(\boldsymbol{x}) = \|\boldsymbol{v} - \boldsymbol{b}\|^2 - \|\boldsymbol{v} + \boldsymbol{L}\boldsymbol{d}\|^2 \tag{46}$$

$$= -2\boldsymbol{v}^{\top}(\boldsymbol{b} + \boldsymbol{L}\boldsymbol{d}) + \|\boldsymbol{b}\|^2 - \|\boldsymbol{L}\boldsymbol{d}\|^2 \tag{47}$$

Because $\boldsymbol{v}$ and $\boldsymbol{x}$ vary independently and the equation is true for any pair of these points, both sides of the equation need to be independent of their respective variables. This is true only if $\boldsymbol{b} = -\boldsymbol{L}\boldsymbol{d}$. Hence, it follows that

$$c'(\boldsymbol{x}) = 0 \quad \text{and} \quad \boldsymbol{d} = -\boldsymbol{L}^{-1}\boldsymbol{b}. \tag{48}$$

Inserting $\boldsymbol{d}$ into Eq. 40 gives the final solution,

$$\boldsymbol{f}(\boldsymbol{x}) = \boldsymbol{L}^{-1}\hat{\boldsymbol{f}}(\boldsymbol{L}\boldsymbol{x}+\boldsymbol{b}) - \boldsymbol{L}^{-1}\boldsymbol{b}. \tag{49}$$

Solving for $\hat{\boldsymbol{f}}$ gives us

$$\hat{\boldsymbol{f}}(\boldsymbol{L}\boldsymbol{x}+\boldsymbol{b}) = \boldsymbol{L}\boldsymbol{f}(\boldsymbol{x}) + \boldsymbol{b} \tag{50}$$

$$\hat{\boldsymbol{f}}(\boldsymbol{x}) = \boldsymbol{L}\boldsymbol{f}(\boldsymbol{L}^{-1}(\boldsymbol{x} - \boldsymbol{b})) + \boldsymbol{b} = (\boldsymbol{r} \circ \boldsymbol{f} \circ \boldsymbol{r}^{-1})(\boldsymbol{x}) \tag{51}$$

which concludes the proof. $\qquad\square$

## B KERNEL DENSITY ESTIMATE CORRECTION

Theorem 1 requires to include a "potential function" $\alpha$ into our model. In this section, we discuss how this function can be approximated by a kernel density estimate (KDE) in practice. The KDE intuitively corrects for the case of non-uniform marginal distributions. Correcting with $\alpha$ overcomes the limitation of requiring a uniform marginal distribution discussed before (Zimmermann et al., 2021). While other solutions have been discussed, such as training a separate MLP (Matthes et al., 2023), the KDE solution discussed below is conceptually simpler and non-parametric.

For the models considered in the main paper, we considered representation learning in Euclidean space, while Appendix D contains some additional experiments for the very common case of training embeddings on the hypersphere. For both cases, we can parameterize appropriate KDEs.

For the Euclidean case, we use the KDE based on the squared Euclidean norm,

$$\hat{q}(\boldsymbol{x}) = \frac{1}{\epsilon M} \sum_{i=1}^{M} \exp\left(-\frac{\|\boldsymbol{x} - \boldsymbol{x}_i\|^2}{\epsilon}\right), \quad \boldsymbol{x}_i \sim q(\boldsymbol{x}). \tag{52}$$

We note that in the limit $\epsilon \to 0$, $M \to \infty$, this estimate converges to the correct distribution, $\hat{q}(\boldsymbol{x}) \to q(\boldsymbol{x})$. This is also the case used in Theorem 1. However, this estimate depends on the ground truth latents $\boldsymbol{x}_i$, which are not accessible during training. Hence, we need to find an expression that depends on the observable data. We leverage the feature encoder $\boldsymbol{h}$ to express the estimator as

$$\hat{q}_{\boldsymbol{h}}(\boldsymbol{y}) = \frac{1}{\epsilon M} \sum_{i=1}^{M} \exp\left(-\frac{\|\boldsymbol{h}(\boldsymbol{y}) - \boldsymbol{h}(\boldsymbol{y}_i)\|^2}{\epsilon}\right), \quad \boldsymbol{y}_i \sim q(\boldsymbol{y}). \tag{53}$$

We can express this estimator in terms of the final solution, $\boldsymbol{r}(\boldsymbol{x}) = \boldsymbol{h}(\boldsymbol{g}(\boldsymbol{x})) = \boldsymbol{Q}\boldsymbol{\Sigma}_u^{-1/2}\boldsymbol{x} + \boldsymbol{b}$ in the theorem. If we express the solution in terms of the ground truth latents again, the orthogonal matrix $\boldsymbol{Q}$ vanishes and we obtain

$$\hat{q}_{\boldsymbol{h}}(\boldsymbol{x}) = \frac{1}{\epsilon M} \sum_{i=1}^{M} \exp\left(-\frac{\|\boldsymbol{\Sigma}_u^{-1/2}(\boldsymbol{x} - \boldsymbol{x}_i)\|^2}{\epsilon}\right). \quad \boldsymbol{y}_i \sim q(\boldsymbol{y}) \tag{54}$$

This corresponds to a KDE using a Mahalanobis distance with covariance matrix $\boldsymbol{\Sigma}_u$, which is a valid KDE of $q$.

We can derive a similar argument when computing embeddings and dynamics on the hypersphere. When, a von Mises-Fisher distribution is suitable to express the KDE, and we obtain

$$\hat{q}(\boldsymbol{x}) = \frac{C_p(\kappa)}{M} \sum_{i=1}^{M} \exp(\kappa \boldsymbol{x}^\top \boldsymbol{x}_i), \quad \boldsymbol{x}_i \sim q(\boldsymbol{x}) \tag{55}$$

where $C_p(\kappa)$ is the normalization constant of the von Mises-Fisher distribution. This again approaches the correct data distribution for $\hat{q} \to q$ as $M, \kappa \to \infty$. Following the same arguments above, but

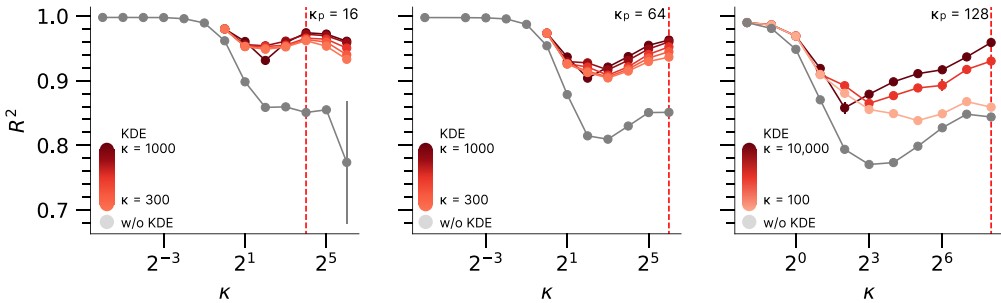

Figure 8: Introducing KDE into the loss allows to compensate for non-uniform marginal distribution. We show performance in terms of $R^2$ across datasets with increasingly non-uniform marginal. We replicate the data-generating process and experimental setup performed by Zimmermann et al. (2021, Figure 2).

using $\boldsymbol{r}(\boldsymbol{x}) = \boldsymbol{h}(\boldsymbol{g}(\boldsymbol{x})) = \boldsymbol{Q}\boldsymbol{x}$ as the indeterminacy on the hypersphere, we can express this in terms of the ground truth latents,

$$\hat{q}_{\boldsymbol{h}}(\boldsymbol{x}) = \frac{C_p(\kappa)}{N} \sum_{i=1}^{M} \exp\left(\kappa \boldsymbol{r}(\boldsymbol{x})^\top \boldsymbol{r}(\boldsymbol{x}_i)\right), \tag{56}$$

$$= \frac{C_p(\kappa)}{N} \sum_{i=1}^{M} \exp\left(\kappa \boldsymbol{x}^\top \boldsymbol{x}_i\right), \tag{57}$$

which is again a valid KDE.

It is interesting to consider the effect of the KDE on the loss function. Inserting $\psi \leftarrow \psi - \log \hat{q}$ into the loss function yields

$$-\log p_\psi(\boldsymbol{x}|\boldsymbol{x}^+, N) = -(\psi(\boldsymbol{x}_i, \boldsymbol{x}_i^+) - \log \hat{q}(\boldsymbol{x}_i^+)) + \log \sum_{i=1}^{N} e^{\psi(\boldsymbol{x}_i, \boldsymbol{x}_j^-) - \log \hat{q}(\boldsymbol{x}_j^-)}, \tag{58}$$

$$= -\psi(\boldsymbol{x}_i, \boldsymbol{x}_i^+) + \log \hat{q}(\boldsymbol{x}_i^+) + \log \sum_{i=1}^{N} \frac{1}{\hat{q}(\boldsymbol{x}_j^-)} e^{\psi(\boldsymbol{x}_i, \boldsymbol{x}_j^-)}, \tag{59}$$

$$= -\psi(\boldsymbol{x}_i, \boldsymbol{x}_i^+) + \log \sum_{i=1}^{N} \frac{\hat{q}_h(\boldsymbol{x}_i^+)}{\hat{q}_h(\boldsymbol{x}_j^-)} e^{\psi(\boldsymbol{x}_i, \boldsymbol{x}_j^-)}, \tag{60}$$

$$= -\psi(\boldsymbol{x}_i, \boldsymbol{x}_i^+) + \log \sum_{i=1}^{N} w_h(\boldsymbol{x}_i^+, \boldsymbol{x}_j^-) e^{\psi(\boldsymbol{x}_i, \boldsymbol{x}_j^-)} \tag{61}$$

with the importance weights $w_h(\boldsymbol{x}_i^+, \boldsymbol{x}_j^-) = \frac{\hat{q}_h(\boldsymbol{x}_i^+)}{\hat{q}_h(\boldsymbol{x}_j^-)}$. Intuitively, this means that the negative examples are re-weighted according to the density ratio between the current positive and each negative sample.

**Empirical motivation.** Figure 8 shows preliminary results on applying this KDE correction to contrastive learning models. We followed the setting from Zimmermann et al. (2021) and re-produced the experiment reported in Fig. 2 in their paper. We use 3D latents, a 4-layer MLP as non-linear mixing function with a final projection layer to 50D observed data. The reference, positive and negative distributions are all vMFs parameterized according to $\kappa$ (x-axis) in the case of the reference and negative distribution and $\kappa_p$ for the positive distribution.

The grey curve shows the decline in empirical identifiability ($R^2$) as the uniformity assumption is violated by an increasing concentration $\kappa$ (x-axis). Applying a KDE correction to the data resulted in substantially improved performance (red lines).

However, when testing the method directly on the dynamical systems considered in the paper, we did not found a substantial improvement in performance. One hypothesis for this is that the distribution of points on the data manifold (not necessarily the whole $\mathbb{R}^d$ is already sufficiently uniform. Hence, while the theory requires inclusion of the KDE term (and it did not degrade results), we suggest to drop this computationally expensive term when applying the method on real-world datasets that are approximately uniform.

## C  ADDITIONAL RELATED WORK

**Contrastive learning.** An influential and conceptual motivation for our work is Contrastive Predictive Coding (CPC) (Oord et al., 2018) which uses the InfoNCE loss with an additional non-linear projection head implemented as an RNN to aggregate information from multiple time steps. Then, an affine projection is used for multiple forward prediction steps. However, contrary to our approach, the "dynamics model" is not explicitly parameterized, limiting its interpretability. Similar frameworks have been successfully applied across various domains, including audio, vision, and language, giving rise to applications such as wav2vec (Schneider et al., 2019), time contrastive networks for video (TCN; Sermanet et al., 2018) or CPCv2 (Henaff, 2020). An interesting parallel to DLC+LDS in image representation learning is AnInfoNCE (Rusak et al., 2024) which extends a SimCLR-like (Chen et al., 2020b) training setup by an additional learnable diagonal matrix in its similarity function.

**Non-Contrastive learning.** Models such as data2vec (Baevski et al., 2022) and JEPA (Assran et al., 2023) learns a representation by trying to predict missing information in latent space, using an MSE loss. JEPA uses asymmetric encoders, and on top a predictor model in latent space parameterized by a neural net. However, these approaches do not provide any identifiability guarantees.

**System identification.** In system identification, a problem closely related to the one addressed in this work is known as "nonlinear system identification. Widely used algorithms for this problem include Extended Kalman Filter (EKM) (McGee & Schmidt, 1985) and Nonlinear Autoregressive Moving Average with Exogenous inputs (NARMAX) (Chen & Billings, 1989). EKF is based on linearizing $g$ and $f$ using a first-order Taylor-series approximation and then apply the Kalman Filter (KF) to the linearized functions. NARMAX, on the other hand, typically employs a power-form polynomial representation to model the non-linearities. In neuroscience, practical (generative algorithms) include systems modeling linear dynamics (fLDS; Gao et al., 2016) or non-linear dynamics modelled by RNNs (LFADS; Pandarinath et al., 2018). Hurwitz et al. (2021) provide a detailed summary of additional algorithms.

**Nonlinear ICA.** The field of Nonlinear ICA has recently provided identifiability results for identifying latent variables, usually employing auxiliary variables such as class labels or time information (Hyvarinen & Morioka, 2016; 2017; Hyvarinen et al., 2019; Khemakhem et al., 2020; Sorrenson et al., 2020). In the case of time series data, Time Contrastive Learning (TCL) (Hyvarinen & Morioka, 2016) uses a contrastive loss to predict the segment-ID of multivariate time-series which was shown to perform Non-linear ICA. Permutation Contrastive Learning (PCL) (Hyvarinen & Morioka, 2017) permutes the time series and aims to distinguish positive and negative pairs.

**Temporal causal representation learning.** In Nonlinear ICA, the factors are assumed to be *independent*, subject to some indeterminacy in the original latent variables. However, this approach encounters challenges when the latent variables have time-delayed causal relationships. Approaches like LEAP (Sorrenson et al., 2020) and TDLR (Yao et al., 2021) address these challenges in both stationary and non-stationary environments, even when the transition function's parametric form is unknown. CaRING (Yao et al., 2022) extends these results to cases where the mixing function is non-invertible. Lastly, CITRIS (Lippe et al., 2022) introduces intervention target information to enhance the identification of latent causal factors. In this work, we do not aim to estimate the temporal causal graph. Instead, we focus on estimating the dynamics model $f$ using an interpretable and explicitly parameterized dynamics model (e.g. $\nabla$-SLDS) which can later be analyzed for applications such as scientific discovery.

**Switching Linear Dynamical Systems.** Several papers propose methods to infer SLDSs (Ackerson & Fu, 1970; Chang & Athans, 1978; Ghahramani & Hinton, 2000), leading to a variety of extensions and variants. For example, Recurrent SLDSs (Linderman et al., 2017; Dai et al., 2022) address state-dependent switching by changing the switch transition distribution to $p(y_t|y_{t-1}, x_{t-1})$, allowing for more flexible dependencies on previous states. Another extension, Explicit duration SLDS introduces additional latent variables to model the distribution of switch durations explicitly (Chiappa et al., 2014). Some approaches relax the assumption of linear dynamics, such as in the case of SNLDS and RSSSM, where the dynamics model is assumed to be nonlinear (Dong et al., 2020; Chow & Zhang, 2013). In the context of Nonlinear Independent Component Analysis (ICA), recent extensions include structured data generating processes (e.g., SNICA; Hälvä et al., 2021) which were shown to be useful for the inference of switching dynamics. In this vein, Balsells-Rodas et al. (2023) proposed additional identifiability theory for the switching case. Other approaches, based on Neural Ordinary Differential Equations (Neural ODEs; Chen et al., 2020a; Shi & Morris, 2021), or methods aimed at

discovering switching dynamics within recurrent neural networks (Smith et al., 2021), also present interesting avenues for modeling switching dynamics.

**Deep state-space models.** Recently, (deep) state-space models (SSMs) such as S4 or Mamba (Gu et al., 2021; Gu & Dao, 2023) have emerged as a promising architecture. These models are particularly well-suited for capturing long-range dependencies, making them an attractive choice for sequence modeling tasks.

**Symbolic Regression.** An alternative approach to modeling dynamical systems is the use of symbolic regression, which aims to directly infer explicit symbolic mathematical expressions governing the underlying dynamical laws. Examples include Sparse Identification of Nonlinear Dynamics (SINdy; Brunton et al., 2016), as well as more recent transformer-based models (Kamienny et al., 2022; d'Ascoli et al., 2023), which have demonstrated promise in discovering interpretable representations of dynamical systems.

# D  VON MISES–FISHER (vMF) CONDITIONAL DISTRIBUTIONS

In the main paper, we have shown experimental results that verify Theorem 1 in the case of Normal distributed positive conditional distribution and using the Euclidean distance. This approach has allowed for modeling latents and their dynamics in Euclidean space, which we argue is the most practical setting to apply DCL in. However, self-supervised learning methods and especially contrastive learning have commonly been applied to produce representations on the hypersphere and using the dot-product distance (Oord et al., 2018; Schneider et al., 2023; Wang & Isola, 2020; Zimmermann et al., 2021; Chen et al., 2020b).

Here we validate empirically that Theorem 1 equally holds under the assumption of vMF conditional distributions and using the dot-product distance $\phi(\boldsymbol{x}, \boldsymbol{y}) = \boldsymbol{x}^\top \boldsymbol{y}$ as part of the loss. We run experiments as in Table 1 for the case where the true dynamics model $\boldsymbol{f}$ is a linear dynamical system. Additionally, we vary the setting similar to Figure 4 to show increasing $\Delta t$ (angular velocity).

We compare:

- **DCL (ours)** – with linear dynamics: $\hat{\boldsymbol{f}}(\boldsymbol{x}) = \hat{\boldsymbol{A}}\boldsymbol{x}$.

- **GTD** – the ground-truth dynamics model (LDS) $\hat{\boldsymbol{f}}(\boldsymbol{x}) = \boldsymbol{A}\boldsymbol{x}$.

- **No dynamics** – the baseline setting we use throughout the paper $\hat{\boldsymbol{f}}(\boldsymbol{x}) = \boldsymbol{x}$.

- **Asymmetric** – a variation on the baseline setting that uses asymmetric encoders (one for reference, one for positive or negative) which would be a possible fix of Corollary 1. We can obtain this setting by skipping the explicit dynamics modeling, and defining two encoders $\boldsymbol{h}_1, \boldsymbol{h}_2$ which relate as follows: $\boldsymbol{h} \circ \boldsymbol{f} := \boldsymbol{h}_1, \boldsymbol{h} := \boldsymbol{h}_2$.

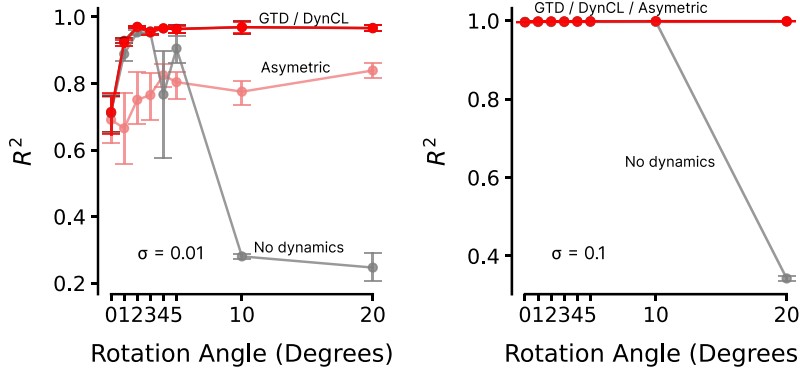

Figure 9: Our findings from Table 1 hold equally under vMF noise distribution when using LDS ground truth dynamics. We show empirical identifiability of the latents in terms of $R^2$ under varying a) angles of the rotation dynamics i.e. angular velocity $\Delta t$ (x-axis) and b) the magnitude of the dynamics noise $\sigma$ (panels, left: low noise, right: high noise)

Similar to our results for the Euclidean case (Table 1), in Figure 9 we show results that experimentally verify Theorem 1 for latent dynamics on hypersphere and using vMF as conditional distribution. Both for low (left panel) and high (right panel) variance of the conditional distribution we can see that DCL effectively identifies the ground truth latents on par with the oracle (GTD) model performance. On the other hand, the baseline, standard time contrastive learning without dynamics, can not identify the ground truth latents with underlying linear dynamics as predicted by Corollary 1. This prediction is only violated in the case where the variance of the noise distribution is high enough, such that the noise dominates the changes introduced by the actual dynamics. This is the case for dynamics with rotations up to 4 degrees for $\sigma = 0.01$ and angles up to 10 degrees for $\sigma = 0.1$.

# E ADDITIONAL PLOTS FOR SLDS

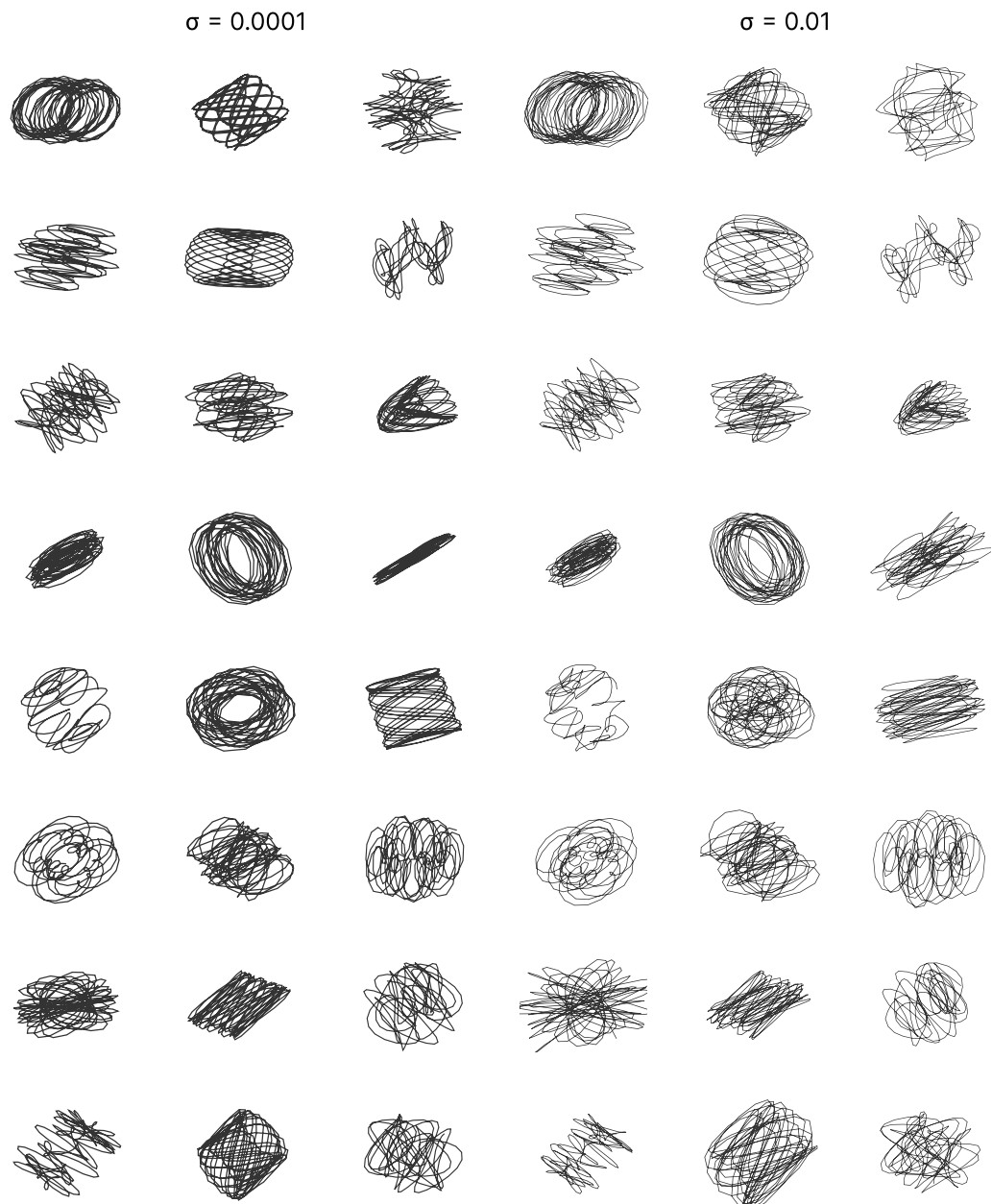

Figure 10: Visualizations of 6D linear dynamical systems at $\sigma = 0.0001$ (left) and $\sigma = 0.01$ for 10 degree rotations. These systems are used in our SLDS experiments.

## F GENERALIZATION – TRAIN- VS. TEST SET

In the main paper, all metrics are evaluated using the full training dataset of the respective experiment. We argue that this is sufficient for showing the efficacy of our model and verifying claims from the theory in section 3 because a) in self-supervised learning, the model learns generalizable representations through pretext tasks, making overfitting less of a concern; b) the metrics we are interested in are about uncovering the true underlying latent representation and dynamics of the available training data, not of new data; and c) most importantly, we ensured that the training dataset is large enough to approximate the full data distribution.

Nonetheless, here we show a series of control experiments to re-evaluate models on new and unseen data. We do so by following the same data generating process of the given experiment (same dynamics model and mixing function) and sample completely new trials (10% of the number of trials of the training dataset). Every new trial starts at a random new starting point, with a randomly sampled new mode sequence and regenerated with different seeds for the dynamics noise.

First, we re-evaluated every experiment of Figure 6 on the test dataset generated as described above and show these results in Figure 11. Comparing those results to the train dataset version of Figure 6 shows that there is almost no difference in the performance (with regard to identifiability and systems identification). The difference are so small, that visually comparing the results almost becomes impractical, so we additionally provide the exact numbers of the first panel (variations on the number of samples per trial) in Table 2.

Finally, to qualitatively and quantitatively show the difference between the train and test datasets we provide a) depictions of the ground truth latents of 5 random test trials and their closest possible matching trial from the training set in Figure 12 and b) a distribution of the distances (in terms of $R^2$ between the data from the test and train trials) between all test trials of one of a random test set and their closest trial from the training dataset in Figure 13.

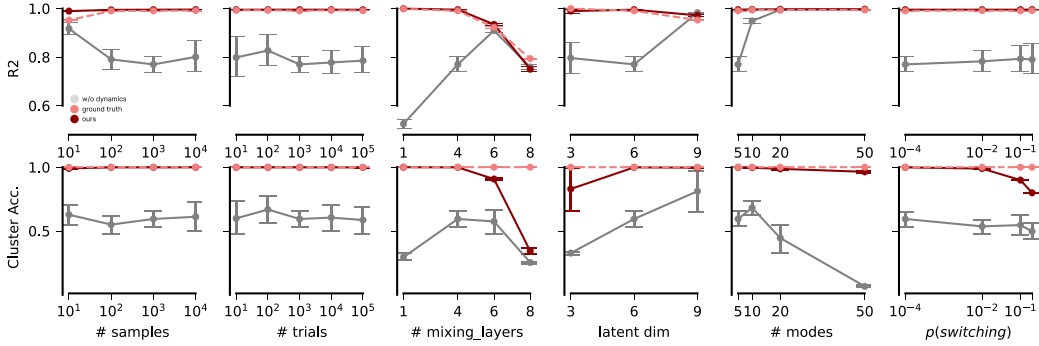

Figure 11: Same as Figure 6 (Variations and ablations for SLDS), but re-evaluated on a newly generated test data with different starting points.

Table 2: A detailed view on the #samples panel from Figure 6 and 11 showing the difference between train and test set evaluation.

| | DCL (ours) | | CL w/o dynamics | | CL w/ ground truth dynamics | |
| | $R^2$ (train) | $R^2$ (test) | $R^2$ (train) | $R^2$ (test) | $R^2$ (train) | $R^2$ (test) |
| # samples | | | | | | |
| --- | --- | --- | --- | --- | --- | --- |
| 10 | $0.991 \pm 0.00137$ | $0.989 \pm 0.00172$ | $0.923 \pm 0.03703$ | $0.917 \pm 0.04000$ | $0.954 \pm 0.00397$ | $0.952 \pm 0.00442$ |
| 100 | $0.995 \pm 0.00106$ | $0.994 \pm 0.00108$ | $0.786 \pm 0.06794$ | $0.791 \pm 0.06931$ | $0.990 \pm 0.00107$ | $0.990 \pm 0.00099$ |
| 1000 | $0.995 \pm 0.00074$ | $0.995 \pm 0.00078$ | $0.765 \pm 0.06671$ | $0.770 \pm 0.06973$ | $0.990 \pm 0.00507$ | $0.990 \pm 0.00511$ |
| 10000 | $0.996 \pm 0.00046$ | $0.996 \pm 0.00044$ | $0.694 \pm 0.06937$ | $0.801 \pm 0.10568$ | $0.991 \pm 0.00509$ | $0.991 \pm 0.00425$ |

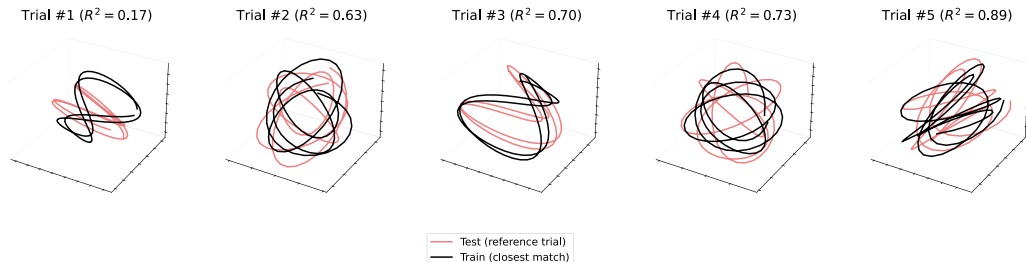

Figure 12: Ground truth latents from five random trials of the testsets for Figure 11 and their closest match within the corresponding trainset. The closest match is evaluated by computing the $R^2$-Score between a given trial from the testset and every possible trial.

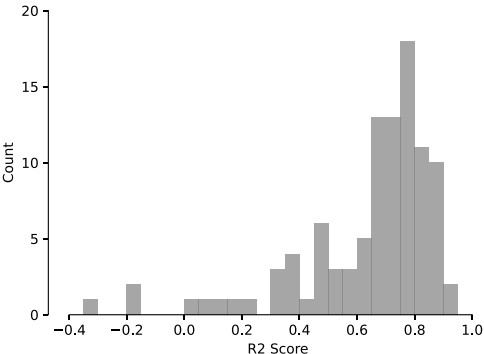

Figure 13: Histogram of all $R^2$-Scores between every trial from the testset and its closest possible match from the trainset as shown in Figure 12.

# G  VARIATIONS AND ADDITIONAL BASELINES FOR THE $\mathrm{Dyn}R^2$ METRIC

## G.1  METHOD

As an addition to Table 1, we analyse the $\mathrm{Dyn}R^2$ in more detail. In Table 3 we show variants for the metric. Firstly, we modify the number of forward prediction steps,

$$\boldsymbol{f}^n(\boldsymbol{x}) := (\underbrace{\boldsymbol{f} \circ \cdots \circ \boldsymbol{f}}_{n \text{ times}})(\boldsymbol{x}) \tag{62}$$

and respectively for $\hat{\boldsymbol{f}}^n$ in relation to $\hat{\boldsymbol{f}}$. We then consider two variants of Eq. 12. Firstly, we perform multiple forward predictions ($n > 1$) and compare the resulting embeddings:

$$\text{r2\_score}(\hat{\boldsymbol{f}}^n(\hat{\boldsymbol{x}}), \boldsymbol{L}\boldsymbol{f}^n(\boldsymbol{L}'\hat{\boldsymbol{x}} + \boldsymbol{b}') + \boldsymbol{b}). \tag{63}$$

A rationale for this metric is that the prediction task becomes increasingly difficult with an increasing number of time steps, and errors accumulate faster.

Secondly, as an additional control, we replace $\hat{\boldsymbol{f}}$ with the identity, and compute

$$\text{r2\_score}(\hat{\boldsymbol{x}}, \boldsymbol{L}\boldsymbol{f}^n(\boldsymbol{L}'\hat{\boldsymbol{x}} + \boldsymbol{b}') + \boldsymbol{b}). \tag{64}$$

This metric can be regarded as a naiive baseline/control for comparing performance of the dynamics model. If the $\mathrm{dyn}R^2$ is not significantly larger than this value, we cannot conclude to have obtained meaningful dynamics.

For the lower part of Table 1, we report the resulting metrics in Table 3, setting the number of forward steps $n$ to 1 or 10, and using either the original metric (Eq. 62), or the control (Eq. 63).

## G.2  RESULTS

For the SLDS system, we can corroborate our results further: the baseline model obtains a $\mathrm{dyn}R^2$ of around 85% for single step prediction, both for the original and control metric. Our $\nabla$-SLDS model and the ground truth dynamical model obtain over 99.9% well above the level of the control metric which remains at around 95%. The high value of the control metric is due to the small change introduced by a single time step, and should be considered when using and interpreting the metric. If more steps are performed, the performance of the $\nabla$-SLDS model drops to about 95.5% vs. chance level for the control metric, again highlighting the high performance of our model, but also the room for improvement, as the oracle model stays at above 99% as expected.

For the Lorenz system, we do not see a substantial difference between original $\mathrm{dyn}R^2$ metric and $\mathrm{dyn}R^2$ control for any of the considered algorithms. Yet, as noted in the main paper, $\nabla$-SLDS is the only dynamics model that gets a high $R^2$ of 94.08%, vs. the lower 81.20% for a single LDS model, or 40.99% for the baseline model. In other words, while DCL with the $\nabla$-SLDS dynamics model falls short of identifying the true underlying dynamics for this non-linear chaotic system, without DCL we wouldn't even identify the latents. We leave optimizing the parameterization of the dynamics model to identify non-linear chaotic systems for future work.

Table 3: Extended metrics for dynamics models including additional variations on the $dynR^2$ metric where *Control* is replacing $\hat{\boldsymbol{f}}$ with the identity and *10 Steps* is applying $\hat{\boldsymbol{f}}$ (and $\boldsymbol{f}$) 10 times, i.e., predicting 10 steps forward instead of only one step as is done in the *Original* version.

| data | | model | | % dyn$R^2$, n=1 step | | % dyn$R^2$, n=10 steps | |
| $\boldsymbol{f}$ | $p(\boldsymbol{\varepsilon})$ | $\hat{\boldsymbol{f}}$ | identifiable | Original | Control | Original | Control |
|---|---|---|---|---|---|---|---|
| SLDS | Normal | identity | ✗ | $85.47 \pm 8.07$ | $84.54 \pm 7.31$ | $2.78 \pm 9.34$ | $-58.62 \pm 7.22$ |
| SLDS | Normal | $\nabla$-SLDS | (✓) | $99.93 \pm 0.01$ | $95.15 \pm 0.68$ | $95.53 \pm 0.47$ | $-124.65 \pm 6.97$ |
| SLDS | Normal | GT | (✓) | $99.97 \pm 0.00$ | $94.94 \pm 0.68$ | $99.36 \pm 0.18$ | $-129.32 \pm 6.95$ |
| Lorenz | Normal | identity | ✗ | $27.02 \pm 8.72$ | $27.17 \pm 8.74$ | $22.87 \pm 7.13$ | $24.85 \pm 7.14$ |
| Lorenz | Normal | LDS | ✗ | $80.30 \pm 14.13$ | $82.98 \pm 12.64$ | $-13.07 \pm 41.03$ | $42.08 \pm 26.14$ |
| Lorenz | Normal | $\nabla$-SLDS | (✓) | $93.91 \pm 5.32$ | $93.70 \pm 5.11$ | $34.48 \pm 6.47$ | $55.75 \pm 6.01$ |

## H  NON-INJECTIVE MIXING FUNCTIONS

Our identifiability guarantees (Theorem 1, Def. 5) require an injective mixing function $\boldsymbol{g}(\boldsymbol{x}_t) = \boldsymbol{y}_t$. On the first glance, this clashes with common requirements in system identification under *partial observability*. Specifically, let us consider a system of the form:

$$
\begin{aligned}
\boldsymbol{x}_{t+1} &= \boldsymbol{f}(\boldsymbol{x}_t) + \boldsymbol{\varepsilon}_t = \boldsymbol{A}\boldsymbol{x}_t + \boldsymbol{\varepsilon}_t \\
\boldsymbol{y}_t &= \boldsymbol{C}\boldsymbol{x}_t = \boldsymbol{g}(\boldsymbol{x}_t),
\end{aligned}
\tag{65}
$$

where $\boldsymbol{x}_t \in \mathbb{R}^n$, $\boldsymbol{y}_t \in \mathbb{R}^m$, $\boldsymbol{C} \in \mathbb{R}^{m \times n}$ with $n > m$ is a non-square matrix that projects the states of the dynamical system into a lower dimensional space. Similarly, we may have $n \leq m$ with $\text{rank}(\boldsymbol{C}) < n$. In those cases, $\boldsymbol{C}$ and hence $\boldsymbol{g}$ is non-invertible, and naively, the injectivity assumptions of Def. 5 would fail to hold.

However, in practice this issue can be tackled through the use of a time-delay embedding. Specifically, we consider the following reformulation of the system:

$$
\tilde{\boldsymbol{y}}_t^i := \begin{bmatrix} \boldsymbol{y}_{t-\tau} \\ \boldsymbol{y}_{t-\tau+1} \\ \boldsymbol{y}_{t-\tau+2} \\ \vdots \\ \boldsymbol{y}_t \end{bmatrix} = \underbrace{\begin{bmatrix} \boldsymbol{C} \\ \boldsymbol{C}\boldsymbol{A} \\ \boldsymbol{C}\boldsymbol{A}^2 \\ \vdots \\ \boldsymbol{C}\boldsymbol{A}^\tau \end{bmatrix}}_{\boldsymbol{O}} \boldsymbol{x}_{t-\tau} + \begin{bmatrix} \boldsymbol{0} \\ \boldsymbol{\nu}_t^1 \\ \boldsymbol{\nu}_t^2 \\ \vdots \\ \boldsymbol{\nu}_t^\tau \end{bmatrix}
\tag{66}
$$

$$
\text{where } \boldsymbol{\nu}_t^\tau := \boldsymbol{C} \sum_{i=0}^{\tau-1} \boldsymbol{A}^i \boldsymbol{\varepsilon}_{t-i}.
\tag{67}
$$

For a sufficiently large time lag $\tau$, the linear map $\tilde{\boldsymbol{g}}(\boldsymbol{x}_{t-\tau}) = \boldsymbol{O}\boldsymbol{x}_{t-\tau} = \tilde{\boldsymbol{y}}_t^i$ will become injective again. This is the case if $\boldsymbol{O}$ is full rank which holds if $\boldsymbol{A}$ is full rank, since $\boldsymbol{f}$ is bijective and therefore $\boldsymbol{A}$ is a full rank square matrix. For example, if $\boldsymbol{C}$ had rank $m$, then using at least $\tau = \frac{n}{m}$ time steps would make $\tilde{\boldsymbol{g}}$ injective and our theoretical guarantees from Theorem 1 would hold, up to the offset introduced by the noise $\boldsymbol{\nu}$.

In practice, the change in latent space between different time steps might be small (especially when the time resolution of the system is very high). A practical way to avoid feeding increasingly large inputs, is to not feed in all time-lags $0 \dots \tau$ into the construction of $\boldsymbol{O}$, but to subselect $k$ time lags $\tau_1, \dots, \tau_k$, with $\tau_1 = 0$ and $\tau_k = \tau$, and instead consider the system

$$
\tilde{\boldsymbol{y}}_t^i := \begin{bmatrix} \boldsymbol{y}_{t-\tau} \\ \boldsymbol{y}_{t-\tau+\tau_2} \\ \vdots \\ \boldsymbol{y}_{t-\tau+\tau_{k-1}} \\ \boldsymbol{y}_{t-\tau_n} \end{bmatrix} = \underbrace{\begin{bmatrix} \boldsymbol{C} \\ \boldsymbol{C}\boldsymbol{A}^{\tau_2} \\ \vdots \\ \boldsymbol{C}\boldsymbol{A}^{\tau_{k-1}} \\ \boldsymbol{C}\boldsymbol{A}^\tau \end{bmatrix}}_{\boldsymbol{O}} \boldsymbol{x}_{t-\tau_1} + \begin{bmatrix} \boldsymbol{0} \\ \boldsymbol{\nu}_t^1 \\ \boldsymbol{\nu}_t^2 \\ \vdots \\ \boldsymbol{\nu}_t^\tau \end{bmatrix}
\tag{68}
$$

This system allows to have a sufficiently large context window (from $t - i_1$ to $t$) to ensure injectivity, while keeping the input dimensionality of the model fixed. Note, when we set $\tau_i = i - 1$ for each $i$, we recover Eq. 66.

Regarding the noise vector $\boldsymbol{\nu}$, Hälvä et al. (2021) recently showed that noisy and noise-free demixing problems can be mapped onto each other. While a full rigorous proof in conjunction with our Theorem 1 is beyond the scope of this work, invoking Theorem 1 of Hälvä et al. (2021) to account for $\boldsymbol{\nu}$ is a promising avenue. Importantly, our empirical validation later on already shows that the model is in practice indeed functioning even in the presence of the noise distribution.

### H.1  EXPERIMENTAL VALIDATION

We validate our theoretical considerations by two sets of experiments as closely related to the LDS setting from Table 1 as possible. We use two types of mixing functions:

$$
\boldsymbol{g}(\boldsymbol{x}) = \boldsymbol{C}_1 \boldsymbol{C}_2 \boldsymbol{x}_t
\tag{69}
$$

$$
\boldsymbol{g}(\boldsymbol{x}) = \boldsymbol{C}_2 \boldsymbol{g}'(\boldsymbol{C}_1 \boldsymbol{x}_t)
\tag{70}
$$

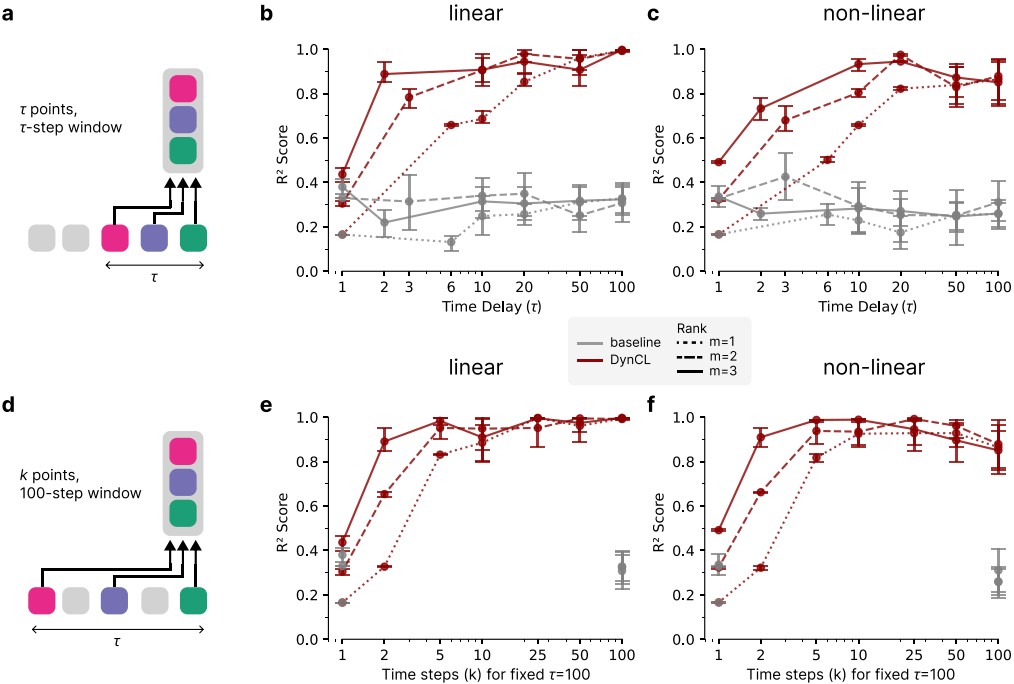

Figure 14: Non-injective mixing functions can be successfully handled by a time-lag embedding. **a**, in the first setting, we pass observations from $\tau$ consecutive time steps into our feature encoder. **b**, empirical identifiability of the latent space ($R^2$) for baseline (no dynamics) vs. DCL (linear dynamics) as we increase $n$ for a linear and **c** non-linear mixing function. **d**, to achieve injective mixing functions through time-lag embeddings, we here include the full 100-step length window, but only pass $k$ equidistantly spaced points within this window of length $\tau$. **e**, empirical identifiability for baseline (no dynamics) vs. DCL (linear dynamics) as we increase the number of points in the context for a fixed $\tau = 100$-step window and **f** for nonlinear mixing.

where $\boldsymbol{C}_1 \in \mathbb{R}^{m \times r}$, $\boldsymbol{C}_2 \in \mathbb{R}^{r \times n}$ are randomly sampled, $\boldsymbol{g} : \mathbb{R}^r \to \mathbb{R}^r$ is a random injective and nonlinear function and $m$ is the observable dimension, $n$ is the latent dimension and $r$ is the matrix rank. In line with the LDS experiments from Table 1, we set $n = 6$ and $m = 50$. The mixing functions from Eq. 69 & 70 are then applied to the latents, including the parameter $r \in \{1, 2, 3\}$ to restrict the rank and thereby dimensionality of the mixing functions to $r$.

We run experiments for different numbers of time steps that get passed to the encoder $\boldsymbol{h}$. In the first setting, we use $k$ consecutive time steps. In this setting, we expect that *more* time steps than the theoretical minimum $n/m$ are required, because the variation between consecutive time steps is limited. To corroborate this hypothesis, we run a second variant where the length of the context window is fixed, and $k$ equidistantly placed points are selected to be fed in the encoder.

As in Table 1, we repeat each experiment 9 times with different dataset and model seeds and report the 95% confidence interval.

## H.2 RESULTS

Generally, we can see from Figure 14 that we can successfully verify our theoretical considerations about weakening the injectivity constraint.

To begin with, we confirm that the default parametrization of DCL with only a single time step $i = 1$ cannot solve the demixing problem. Next, for the theoretical minimum of time steps ($i = 6$ for rank 1, $i = 3$ for rank 2, $i = 2$ for rank 3) we can already observe a considerable improvement of the empirical identifiability for DCL in terms of the $R^2$ Score: $16\% \to 66\%$ for rank 1, $30\% \to 78\%$ for rank 2, $43\% \to 88\%$ for rank 3 (Fig. 14b). As mentioned above, the minimal amount of time steps required for the identifiability guarantees only hold for $\boldsymbol{\nu}_t = 0$ which is not the case in these experiments. As we increase the number of time points to 100, we can see that DCL approaches close to perfect $R^2$ Scores: For linear mixing, we get the best results for $i = 100$ with 99% $R^2$ for

ranks 1, 2, and 3. While for nonlinear mixing, averaged across seeds, we get up to 86% for rank 1 and $i = 100$, 97% for rank 2 and $i = 20$, and 94% for $i = 20$. In contrast, the baseline without a dynamics model, does not benefit at all from the additional time steps.

To further test our intuition about recovering injectivity, we include an experiment where the full 100-step length window is used, but only $n$ equidistantly spaced time steps are passed. In this setup, we much quicker recover acceptable identifability scores for the linear (Fig. 14e) and non-linear mixing settings (Fig. 14f).

# I  DYNAMICAL SYSTEMS WITH CONTROL SIGNAL

We have initially introduced the problem formulation of this paper (Equation 1) as identifying the latent variables $x_t$ and the governing latent dynamics $f$ from the observations $y_t$ for:

$$x_{t+1} = f(x_t) + Bu_t + \varepsilon_t$$
$$y_t = g(x_t) + \nu_t. \tag{71}$$

with control signal $u$, its control or actuator matrix $B$, system noise $\varepsilon$ and observation noise $\nu$.

However, so far we have only explicitly considered autonomous latent dynamics (see Def. 5), which by definition did not include a control input $u_t$. In this section, we show that the DCL framework works equally well in the presence of a control input and verify this empirically.

Being able to include a control signal is crucial for many applications and common practice in systems identification literature. Without adding $u_t$ to the model, the task of identifying dynamics gets substantially harder as the effects of the control signal become entangled with the intrinsic dynamics of the system. Additionally, only identifying the combined dynamics without factoring out the effect of $u_t$ would make the framework less useful as it would not allow predictions in the presence of new/different control inputs.

## I.1  EMPIRICAL VERIFICATION

We extend our existing experiments for linear dynamical systems (LDS) by including a control signal $u_t$ in the data generating process:

$$x_{t+1} = Ax_t + Bu_t + \varepsilon_t \tag{72}$$

We train and evaluate four model variants:

- **Baseline**: Identity dynamics model, using the control input with with a trainable $B$. The dynamics model is fit post-hoc,
- **DCL**: with a LDS dynamics model where $B = 0$,
- **DCL w/ control:** with a LDS dynamics model and trainable $B$.

We choose the control signal $u_t$ to be generated from:

- **Step function**: A composition of a negative and positive step function, starting at random time steps and random magnitudes.
- **Linear Dynamical System**: $u_{t+1} = A_u u_t + \varepsilon_t$, similar to the LDS system used before for latent dynamics.

## I.2  EXPERIMENT DETAILS

We generate three datasets with linear dynamics using a) no control, b) control following another LDS, and c) control following a step function. Each dataset consists of 1000 trials, each trial is 1000 time steps long. The latent linear dynamics have intrinsic rotational dynamics with rotation angles $\theta_i \sim \text{Uniform}[0, 10]$ and control matrix $B \sim \mathcal{N}(\mu = 0.01, \Sigma = I \cdot 0.01)$. The system noise $\varepsilon$ follows a standard normal with $\sigma_\varepsilon = 0.001$. The mixing function $g$ is the same nonlinear mixing function with 4 layers as for Table 1. We use 5-dimensional latents and have 50-dimensional observations. For the control following another LDS, we use the same sampling strategy for the parameters as for the latent dynamics. For the control following a step function, we pick two points $T_1, T_2$ such that the $T_2 - T_1 = 200$ and that the step is centered within the trial including some random offset. We use different controls $u_t$ for every trial. We extend DCL with a parametrization of the linear dynamics model that follows Eq. 72 and includes the control $u_t$ and trainable control matrix $B$. We use the same dynamics model for the baseline. However there, this only affects the post hoc dynamics fitting. We also train DCL with a LDS model that does not include the control input. We train every model for 20k steps and besides that, we use the same hyperparameters as for training the LDS models in Table 1.

Table 4: Empirical identifiability results when including both system noise and a deterministic control signal $\boldsymbol{u}$. The ground truth dynamics $\boldsymbol{f}$ are chosen as an LDS, and the intrinsic dynamics model $\hat{\boldsymbol{f}}$ is either an LDS, or the identity (baseline); $\boldsymbol{Bu}$ indicates whether the dynamics model includes the control or not; $p_\varepsilon(\boldsymbol{\varepsilon})$ is Normal (small $\sigma$). Mean $\pm$ std. are across 3 datasets and 3 experiment repeats.

| Data | Model | | Results | |
|---|---|---|---|---|
| Control ($\boldsymbol{u}$) | $\hat{\boldsymbol{f}}$ | $\hat{\boldsymbol{B}}\boldsymbol{u}$ | %$R^2$ ↑ | %dyn$R^2$ ↑ |
| Step (1D) | identity | ✓ | $91.80 \pm 1.15$ | $87.27 \pm 10.6$ |
| | LDS | ✗ | $98.36 \pm 0.66$ | $99.16 \pm 1.00$ |
| | LDS | ✓ | $98.70 \pm 0.44$ | $99.53 \pm 0.27$ |
| LDS (5D) | identity | ✓ | $74.17 \pm 13.8$ | $76.47 \pm 7.51$ |
| | LDS | ✗ | $98.26 \pm 0.17$ | $99.87 \pm 0.04$ |
| | LDS | ✓ | $98.10 \pm 0.35$ | $99.85 \pm 0.08$ |

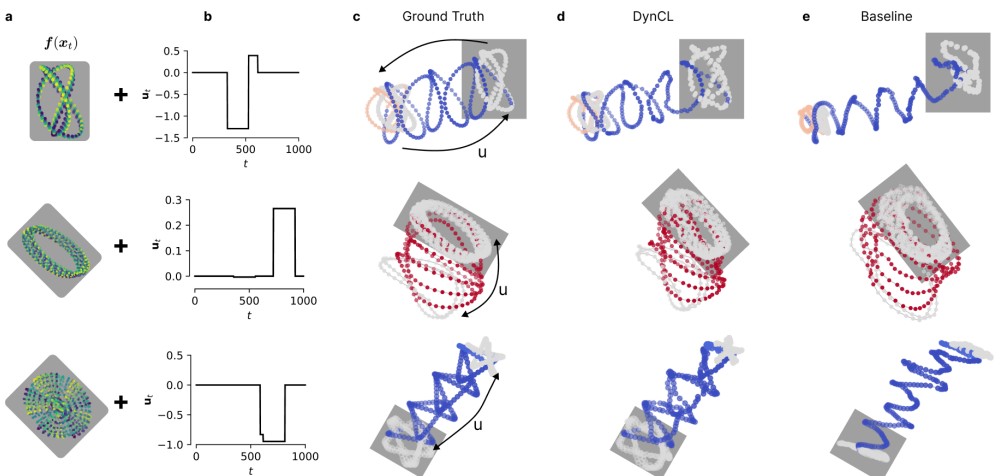

Figure 15: Visualization of dynamics inference in the presence of a control signal $\boldsymbol{u}$. **a** LDS dynamics are complemented by a **b** 1D step function signal, which **c** is projected to 5D using a random matrix $\boldsymbol{B}$ (not shown). The ground truth dynamics then show signatures of the autonomous dynamics when $\boldsymbol{u} = 0$ (gray box), and move through latent space as we apply the step function. **d**, DCL uses a dynamics model and the control input $\boldsymbol{u}$; **e**, the baseline uses only the control input $\boldsymbol{u}$. The three rows show different dynamical systems, each plot is one trial with 1000 steps.

## I.3  RESULTS

As shown in Table 4, when applying a step function for the control signal (Step 1D), DCL is able to identify both the latent space (98.7% $R^2$) and the dynamics (99.5% dyn$R^2$). This result holds even when the control signal $\boldsymbol{u}$ is not used for training the model. However, a baseline with identity dynamics is not able to identify the dynamical system with a substantially lower dyn$R^2$ of 87.3%. Nevertheless, the latent space can be estimated reasonably well, although not perfect (91.8 % $R^2$), most likely because the availability of $\boldsymbol{u}$ converts the de-mixing problem into a supervised learning problem (we have access to $(\boldsymbol{u}, \boldsymbol{y})$ pairs). This is reflected in the visualization in Figure 15: While the baseline (identity dynamics, but using $\boldsymbol{u}$) reasonably estimates the direction of $\boldsymbol{u}$ provided during training, the local dynamics cannot be fitted. We highlighted a gray box denoting the phase where $\boldsymbol{u} = 0$ to facilitate easier comparison.

To test DCL for more complex control signals, we also apply a full 5D LDS as the control signal $\boldsymbol{u}$ (Table 4, LDS 5D). Both latent space estimation and dynamics estimation performs on par with the step setting (>98%), while the baseline fails to estimate either the latent space or the dynamics with $R^2$ values below 80%.

## J    APPLICATION TO REAL-WORLD DATA

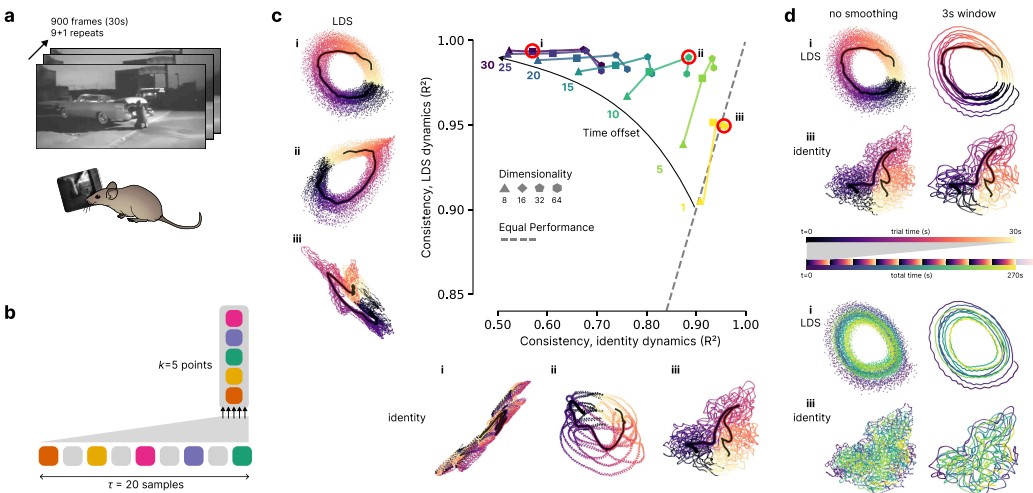

Figure 16: Application to real-world neural data. (a) We leverage the Allen visual coding dataset using calcium imaging data (de Vries et al., 2020). We consider the subset of the data which is comprised of 10 movie repeats with 900 samples each at 30Hz. We hold out the last trial for validation. Mouse icon from scidraw.io (Ann Kennedy), panel adopted from Schneider et al. (2023). (b) We feed multiple time steps to the model as outlined in Appendix H. All experiments concat five equidistant points with a maximum time lag of at the sample locations $(t, t-5, \dots, t-20)$. (c) Quantitative comparison of DCL with identity vs. linear dynamics and qualitative overview of embeddings computed for the hyperparameters that (i) maximize consistency for LDS dynamics, (ii) have high consistency for both models, (iii) maximize consistency for identity dynamics. Train embedding is depicted as points (unsmoothed for LDS, smoothed for identity to show structure), test embedding as smoothed black trajectory. (d) Effect of smoothing applied to the embedding, left column depicts unsmoothed, right column smoothed embedding on train repeats. Upper panel is colored according to trial time, lower panel is colored according to overall time (train portion of data, test only shown above). In all embedding plots, the test-embedding smoothed with a 151-sample filter is overlayed as a black line.

Here we evaluate DCL using a real-world dataset obtained from the Allen Institute (de Vries et al., 2020). The dataset contains recordings from awake, head-fixed mice as they viewed visual stimuli including three movies on a continuous loop. The recordings were collected using 2-photon calcium imaging. Our analysis focuses on paired data from 10 repetitions of "Natural Movie 1" comprised of 900 frames (Fig 16a). This exact dataset was also used by Schneider et al. (2023) and available as `allen-movie-one-ca-VISp-800-*` in the CEBRA software package.

### J.1    METHODS

We train DCL with an LDS dynamics model. As our baseline, we use the identical training setup but with identity dynamics as in our synthetic experiments. We use an MLP with three layers. The number of units for each layer scales with the embedding dimensionality $d$. The last hidden layer has $10d$ units and all previous layers have $30d$ units. We train on batches with 512 samples (reference and positive) and use $2^{14} = 16384$ negative samples. The model is optimized using the Adam optimizer (Kingma, 2014) with learning rate $10^{-4}$. We train DCL for 10k steps using the negative mean squared error as the similarity metric. For both models we also make use of the time-lag embeddings introduced in Appendix H (Figure 14), setting $k = 5$ and $\tau = 20$ (Figure 16b). Additionally, we vary the dimensionality $d \in \{8, 16, 32, 64\}$ of the embeddings. We also vary the time offset $i$ that determines the time step of the positive sample $t_{\text{pos}} = t_{\text{ref}} + i$ relative to the time step of the reference sample and run experiments for $i \in \{1, 5, 10, 15, 20, 25, 30\}$. We train each model 5 times with different initializations. For a shuffle control, we randomize the time dimension within each individual movie repeat (900 frames) and conduct the same training as previously discussed. We train on the neural activity of the first 9 repetitions (8100 samples, 270s) and use the 10th (900 samples, 30s) for evaluation. To evaluate the models, we compute the consistency metric – the $R^2$ metric between the embeddings – used by Schneider et al. (2023) between the five runs of the same model and training setup as a proxy for empirical identifiability. For the consistency calculation, we

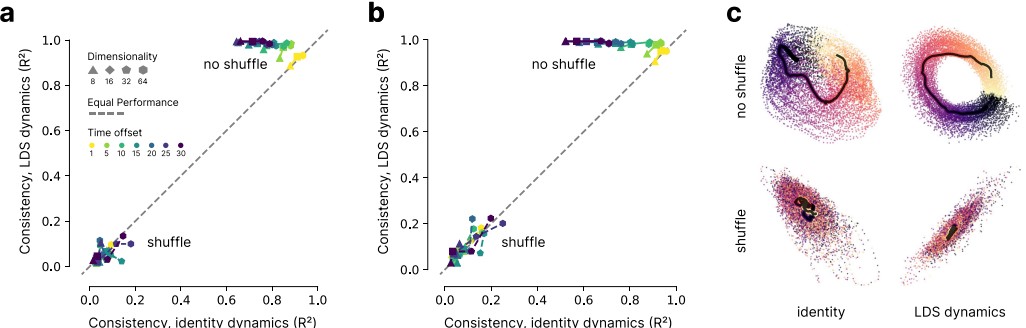

Figure 17: Comparison of consistency with a shuffle control on (a) the train split and (b) the validation split. (c), example embeddings for both identity and LDS dynamics, in shuffle and no shuffle conditions.

fit a linear regression model between all pairs of embeddings across training runs, and quantify the resulting $5 \times 4$ comparisons using the $R^2$. We report the mean $R^2$ on the validation trial across all comparisons.

## J.2 RESULTS

We compare the consistency scores between multiple runs of the same model training for DCL with LDS and our baseline (Figure 16c). Overall, DCL with a linear dynamics model has higher consistency scores compared to training without a dynamics model for all settings except for time offset equal to 1, where results are on par. The positive effect of leveraging dynamics learning increases as we increase the time offset. Across all variations of the time offset, we additionally observe an increase in consistency with an increase in the dimensionality up to 32 dimensions, followed by a drop in consistency for 64-dimensional embeddings.

In addition to the quantitative performance improvements of DCL with LDS, we can also observe more structured embedding spaces for DCL with LDS in Figures 16c and 16d compared to DCL without a dynamics model. The embeddings of DCL with LDS exhibit a clear manifold and follow relatively smooth trajectories, while the embeddings and trajectories of DCL without a dynamics model are considerably more entangled. Considering the color code showing the relative time of each trial, we can also see that DCL with LDS recovers an embedding space in which each trial follows roughly the same circular motion as the other trials.

When removing temporal structure by shuffling (Fig. 17), neither embedding shows non-trivial structure and the consistency metric is low on both the train (panel a) and validation set (panel b).

## J.3 DISCUSSION

We observe the strongest improvements in consistency for those settings in which the time difference between the reference and positive sample is the largest (time offset 30). Under our assumed dynamics model, predicting further ahead in time results in dynamics which differ increasingly from the identity dynamics. In our synthetic data experiments, we already observed a similar effect. The identity dynamics baseline was able to estimate an embedding space when the noise dominated the system dynamics, but failed to estimate the embedding space as the dynamics became more prominent (Fig 5).

Note that our baseline model is similar – but technically not identical – to the CEBRA-time models considered by Schneider et al. (2023) on the same dataset. Our results demonstrate potential improvements through the introduction of the dynamics model: First, we demonstrate that fitting embeddings in Euclidean space using a negative mean squared error as the loss function, vs. the cosine similarity used by Schneider et al. (2023) benefits from the introduction of a dynamics model. Second, the circular, repetitive structure of movie repeats also emerges in this Euclidean space, most clearly in the presence of a dynamics model (Fig. 16c, i, LDS). Third, we found it crucial to include the positive sample in the denominator which stabilizes training in the absence of normalized embeddings.

## K  COMPUTATIONAL REQUIREMENTS

As stated in the main paper, we required 120 GPU days of compute for the experiments we ran for the paper. This does not include the initial period of prototyping and exploration that preceeded the final sweep of experiments. To provide more transparency and more detailed breakdown, we list the number of experiments (= model trainings) run for each table and figure in the paper.

| Result | Number of Experiments |
|---|---|
| Table 1 | 171 |
| Figure 4 (SLDS) | 162 |
| Figure 5 & 7 (Lorenz) | 675 |
| Figure 6 (Ablation) | 648 |
| Figure 8 (KDE) | 1,125 |
| Figure 9 (SLDS with vMF) | 432 |
| Total | 3,213 |

Table 5: Number of experiments per table/figure.

For 120 GPU days this comes out at an average of 53 minutes per experiment that made it into the final paper. However, the actual runtime of an average DCL training is 15-20 minutes for settings equivalent to the experiments in Table 1. This difference to what we report as the overall average compute time per experiment can be attributed to several factors: (1) approximately one-third of experiments involved KDE estimation which required 4-6x longer training times, (2) extensive evaluation and metric computation for debugging and reporting purposes added additional overhead, and (3) additional experimental iterations that did not make it into the final paper but contributed to the total compute time of the final sweep.

## L    ON COMPONENT-WISE VS. LINEAR IDENTIFIABILITY

In the context of non-linear ICA, the mean correlation coefficient (MCC) is frequently reported as a measure of *component-wise identifiability*. In non-linear ICA, it is typically assumed that a set of independent sources $s_1(t), \ldots, s_n(t)$ is passed through a mixing function to arrived at the observable signal (cf. Hyvarinen & Morioka, 2017). In contrast, in our work the sources are not independent, but are conditioned on the previous time step and the passed through a dynamics model. This is conceptually similar to the conditional independent assumption in Hyvarinen et al. (2019) with auxiliary variable $\boldsymbol{f}(\boldsymbol{x})$, but with the distinction that at training time, we do not have $\boldsymbol{u}$ available, only $\boldsymbol{x}$ which requires the use of a dynamics model.

Because of these distinctions in the generation of the latent space, we can generally not expect component-wise identifiability (Theorem 1) but will instead obtain linear identifiability. Related work in non-linear ICA likewise can only provide linear identifiability for a comparable Gaussian case (Hyvarinen et al., 2019; Zimmermann et al., 2021; Schneider et al., 2023). However, if we assume access to the dynamics model (but not the actual latent space), it is possible to reduce ambiguity in the latent space, and assuring component-wise identifiability.

In Table 6 we compare MCC and $\%R^2$ across two datasets (SLDS and Lorenz) to extend Table 1 of the main paper. As expected, we observe a non-perfect MCC score for all models except for the SLDS model which is provided with the ground-truth dynamics. Due to Eq 29 this strenghten the guarantee to component-wise identifiabilily, yielding an MCC of close to 100%.

| Data $\boldsymbol{f}$ | Model $\hat{\boldsymbol{f}}$ | MCC | $\%R^2$ |
|---|---|---|---|
| SLDS | identity | $0.59 \pm 0.06$ | $76.80 \pm 7.40$ |
| SLDS | SLDS | $0.70 \pm 0.05$ | $99.52 \pm 0.05$ |
| SLDS | GT | $1.00 \pm 0.00$ | $99.20 \pm 0.10$ |
| Lorenz | identity | $0.34 \pm 0.07$ | $41.00 \pm 8.57$ |
| Lorenz | LDS | $0.68 \pm 0.14$ | $81.20 \pm 16.9$ |
| Lorenz | SLDS | $0.78 \pm 0.13$ | $94.08 \pm 2.75$ |

Table 6: SLDS and Lorenz dataset from Table 1 with the addition of the MCC metric.

## M    COMPARISON TO ADDITIONAL TIME-SERIES MODELS

The baseline – DCL without a dynamics model – employed in all experiments in the main paper is technically a CEBRA-time (Schneider et al., 2023) model which was originally designed for time-series inference. Note that we use the negative mean squared error as the similarity measure in almost all experiments, and include general improvements for estimation of Euclidean embeddings also in the baseline model (positive sample in the denominator of the InfoNCE loss, additional negative examples). Other modes of CEBRA-time (using a cosine similarity, etc.) were not prominently considered in this work. Other popular contrastive learning methods include time-contrastive learning (TCL; Hyvarinen & Morioka, 2016), permutation contrastive learning (PCL; Hyvarinen & Morioka, 2017). More recently, VAE models designed for time-series analysis and dynamics learning were proposed, such as Temporally Disentangled Representation Learning (TDRL; Yao et al., 2022).

Naturally, these methods propose different data generating processes, and the empirical performance of DCL as well as these comparison methods will strongly depend on whether these assumptions are met. For instance, TCL requires non-stationary independent sources, PCL requires stationary independent sources, and TDRL uses non-parametric transition models with non-Gaussian noise. Hence, in general, it is not possible to fairly compare all these methods as they have different regimes of operation. We still include a comparison of these methods and DCL for dynamics inference.

### M.1    VERIFYING BASELINES

Yao et al. (2022) published a benchmarking setup for these algorithms[3]. We adapted the TDRL codebase minimally and provide a reference implementation for our experiments in our official code release. Our implementation can be diffed against the original TDRL code to highlight the minimal code changes performed for running the benchmarking suite. We leverage this codebase to run verified baseline algorithms on our SLDS dataset. First, we ensure that we can reproduce the results from Yao et al. (2022). We report the results in Table 7 for the "changing" experimental setting.

We perform 5 runs with different seeds for the exact hyperparameter configurations reported in the TDRL codebase. Remaining differences in the in numbers might be attributed to discrepancies between the paper and the code distribution or the choice of different random seeds, as we observed large variances in some of the models. We label these models with "-B" to indicate the *base* configuration provided by the public code base. We report the full results in Table 7.

| | MCC | |
| Model | Reproduced | Reported |
| --- | --- | --- |
| PCL-B | $0.535 \pm 0.030$ | $0.599 \pm 0.041$ |
| TCL-B | $0.367 \pm 0.018$ | $0.399 \pm 0.021$ |
| TDRL-B | $0.910 \pm 0.067$ | $0.958 \pm 0.017$ |

Table 7: Verification on "changing" experiment Setting from Yao et al. (2022)

### M.2    EXPERIMENT DETAILS

Both TCL and TDRL make use of a categorical context variable indicating changes in the distributions of the true latents. To make the comparison to our framework as fair as possible, we choose the SLDS datasets to allow for a similar context variable in form of the mode/state sequence that modulates the switching between linear dynamics. Our dataset is comprised of 5 modes, which corresponds to the same number of categories the models from Table 7 already use.

**Base models.** To be able to apply the baseline models to our SLDS setting, we only change their base configuration in two required ways: We increase the input dimension from 8 to 50 to match the observation produced by the SLDS and reduce the latent dimension from 8 to 6.

**Large models.** Because PCL is the closest match to our existing baseline (CEBRA-time) and our encoder architecture is equal to the baseline architecture, we introduce an additional variant "PCL-L" (L=Large) to match the number of parameters as close as possible. We do so by increasing the hidden dimension of the PCL encoder model from 50 to 160 and reduce the number of layers from 4 to 3,

---

[3]Code: `https://github.com/weirayao/tdrl` (MIT License)

|  | low noise | | high noise | |
| Model | MCC (%) | $R^2$ (%) | MCC (%) | $R^2$ (%) |
| --- | --- | --- | --- | --- |
| TCL-B | $36.07 \pm 2.27$ | $66.56 \pm 3.87$ | $37.21 \pm 3.66$ | $61.75 \pm 12.56$ |
| PCL-B | $68.28 \pm 2.40$ | $91.33 \pm 0.85$ | $66.86 \pm 3.16$ | $77.99 \pm 3.93$ |
| PCL-L | $68.02 \pm 2.77$ | $90.92 \pm 1.16$ | $69.67 \pm 3.86$ | $80.88 \pm 1.38$ |
| TDRL-B | $64.34 \pm 6.01$ | $83.85 \pm 7.78$ | $62.93 \pm 5.37$ | $80.90 \pm 7.82$ |
| TDRL-L | $63.51 \pm 4.87$ | $84.01 \pm 7.98$ | $62.65 \pm 6.29$ | $81.40 \pm 6.42$ |
| CEBRA-time | $59.46 \pm 5.84$ | $76.80 \pm 7.40$ | $65.71 \pm 4.99$ | $98.66 \pm 0.19$ |
| DCL+SLDS | $69.62 \pm 4.78$ | $99.52 \pm 0.05$ | $68.76 \pm 5.05$ | $98.95 \pm 0.08$ |
| DCL+GT SLDS | $99.51 \pm 0.06$ | $99.20 \pm 0.10$ | $98.57 \pm 0.16$ | $97.82 \pm 0.17$ |

Table 8: Baseline results for TCL, PCL, and TDRL models on switching linear dynamics datasets. The low noise setting is equivalent to Table 1. For the high noise setting (low $\Delta t$), we use larger noise and lower rotation angles, setting $\sigma_{\boldsymbol{\varepsilon}} = 0.001$, $\max(\theta_i) = 5$.

effectively increasing the number of parameters by factor 5. Because TDRL can be considered the most promising baseline candidate (beside CEBRA-time) based on the results from table 7, we also double its encoder size from using hidden dimension 128 to 256, resulting in the "TDRL-L" baseline model.

**Dataset.** Leverage two versions of the SLDS dataset used in the main paper. First, we apply it to the exact setting of the SLDS in our Table 1 to compare against our default setting with dynamics noise $\sigma_{\boldsymbol{\varepsilon}} = 0.0001$ and rotation angles $\max(\theta_i) = 10$. Additionally, since our main baseline (CEBRA-time) performed best on datasets with lower $\Delta t$ where the noise dominates over the dynamics, we also compare against an SLDS dataset generated with larger dynamics noise $\sigma_{\boldsymbol{\varepsilon}} = 0.001$ and smaller rotation angles $\max(\theta_i) = 5$ (see Figure 4b). We generate 3 different versions of each dataset using different random seeds and on each dataset we train 3 models with different seeds, resulting in 9 models for each baseline and for each of the two settings. We train every baseline model for 50 epochs or until the training time reaches 8 hours.

**Metrics.** We compute the Mean Correlation Coefficient (MCC) as well as the $R^2$ metric used for Table 1 during training. For the baselines run with the public code base from Yao et al. (2022), we follow their reporting strategy and report the best MCC complemented by the the best $R^2$ achieved at any point during training to make the baseline appear even stronger. For our DCL models and our default baseline, we report the MCC and $R^2$ based on the last model checkpoint as in the main paper.

## M.3 Results

We outline results for all benchmarked algorithms in Table 8.

In the low noise setting, DCL with the SLDS dynamics model achieves an $R^2$ of 99.5%. The next best baseline algorithm is the PCL base model, with a maximum $R^2$ of 91.3%. While both DCL and PCL are learning by contrasting samples across time in the time series, only DCL with the SLDS dynamics model can fully model the ground truth dynamical process. In contrast, the score function in PCL is setup to model variations along *independent* latent dimensions.

Interestingly, PCL outperforms CEBRA-time, which is equivalent to running DCL without a dynamics model (76.8%). This indicates that the score method in PCL (component-wise linear transformations) outperforms the score method in CEBRA-time on this dataset (negative squared Euclidean distance). It would be interesting to combine the scoring method in PCL with the dynamics model in DCL for further improvements on non-linear dynamics settings.

In the high noise setting, DCL with SLDS dynamics achieves comparable performance as CEBRA-time, as outlined in the main paper (99.0% vs. 98.7%). In this setting, PCL performs worse, potentially because the score function in CEBRA-time is better matched to the dominating Gaussian system noise.

In all cases, MCC is not a meaningful metric, with highest scores ranging around 60–70%. An exception is training DCL with the underlying ground truth dynamics system, which achieves component-wise identifiability and an MCC of 99.51%.

