# OpenReview forum: "Self-supervised contrastive learning performs non-linear system identification"
_ICLR.cc/2025/Conference — ICLR 2025 Poster_

### Official Review · Reviewer_qeKY · 2024-11-03

**Soundness:** 3
**Presentation:** 3
**Contribution:** 3
**Rating:** 8
**Confidence:** 3

**Summary:**

This paper focuses on contrastive learning (CL) methods for dynamical systems.
The authors show that under certain assumptions CL performs system identification and can therefore uncover the latent dynamics of the data.
The theoretical findings are applied to switching linear dynamics and non-linear dynamics, and are demonstrated from an empirical point of view using simulated data.

**Strengths:**

1. Overall this is an interesting paper, that gives an insight on why CL techniques are effective for system identification
2. The introduced CL variant is well presented and theoretically grounded, and could inspire further theoretical research and models
3. DynCL can effectively identify latent states and system dynamics in the experiments on simulated data
4. The authors present a good selection of ablation studies to demonstrate the impact of the different modeling/parametric choices

**Weaknesses:**

MORE REALISTIC EXPERIMENTS

As stated by the authors in the limitations of this work, the focus of this paper is only on simulated data. While I understand their point of view, and I also agree that the theoretical contribution/simulated experiments are also valuable by themselves, I fear that the impact of this work in this current state will be more limited than it could be if there was a better demonstration of real world applicability.

You could for example try to apply your method to some of the datasets used in "Discovering State Variables Hidden in Experimental Data" (https://www.cs.columbia.edu/~bchen/neural-state-variables/).

If the above is too challenging to achieve, you should at least try to discuss more in detail what each of you theoretical assumptions means in practice, and what you expect to happen if they are not met in real-world experiments. For example, the fact that $p(u_t)$ is a normal distribution seems quite strict in many applications.

BASELINE

The baseline you use in your experiment seems quite weak, as it does not even use a dynamics model. Have you tried other approaches, for example models doing next-token prediction tasks?




CLARITY

There are several missing definitions/clarifications in the paper that make it a bit harder to follow:
1. N in (3) is not defined
2. "Supp figure" in line 133 is unclear
3. Not sure what "(…)" in line 145 means
4. The name $\nabla$-SLDS is never formally defined
5. In table 1 you have a column called "theory" with different options. What do these option represent exactly?
6. The abbreviation "GT", which I assume stands for ground truth is used in many places but never defined
7. What is $\pi$ in equation (8)?
8. The vMF abbreviation is never defined
9. The DynCL results from Table 1 should be discussed more in depth.

**Questions:**

1. You use the Gumbel-softmax to approximate the argmin. In my experience, the proper turning of the schedule of its temperature parameter is quite challenging. Is it something you noticed as well?

---

### Official Review · Reviewer_3gxR · 2024-11-03

**Soundness:** 3
**Presentation:** 2
**Contribution:** 3
**Rating:** 6
**Confidence:** 3

**Summary:**

The paper "Self-Supervised Contrastive Learning Performs Non-Linear System Identification" explores contrastive learning (CL) for identifying non-linear temporal representations. It presents proof of the identifiability of latent variables up to a linear transformation, removing the requirement for independent noise. The proposed model, DynCL, is validated using synthetic data to support the theoretical findings. Additionally, the authors introduce a model called delta-SLDS, designed to capture switching between linear and non-linear dynamic systems.

**Strengths:**

- The theorem is both interesting and novel, demonstrating that the learned latent variables can be identifiable up to linear transformations even in the absence of independent noise, provided that some other assumptions are met.
- This theorem provides valuable insights into the mechanisms underlying contrastive self-supervised learning methods.
- A lot of ablation studies and visualization experiments are conducted, which makes the paper more convincing.

**Weaknesses:**

- Notations are not clear. For example, in Eq (3), the meanings of $x, x', x''$ should be mentioned in advance.
- In theorem (1) and its proof, the assumption is not aligned with Equation (1-2), where all noise disappears. Further discussion is required.
- The difference in theorem part should be compared with previous works on CL more detailed, since it is a work focusing on theorem. Making the difference more clear will make it more readable.
- In experiment parts
    - Baselines like TCL should be compared
    - By identifiability, some metrics like MCC should be compared even though they are not component-wise identifiable.
- Lots of typos:
    - line 133 (supp figure), line 145 (...), seems unfinished part
    - line 250: tailor -> talyor
    - line 637: Theorem 2 -> Theorem A1
    - footnote of page 13: broken reference
    - line 659, 665: missing reference

**Questions:**

- It is confusing why it requires 120 GPU days on A100 for these synthetic data. Methods like TCL require only dozens of minutes for one synthetic data. What is the detailed model structure with approximate parameter size?

---

> ### Comment · Reviewer_3gxR · 2024-11-26
>
> Thank you for your reply. After reading your reply as well as the other reviewers comments, I decided to raise my score accordingly.

---

> > ### Author Response · Authors · 2024-11-28
> > **Additional benchmarking**
> >
> > Dear reviewer, thank you for re-evaluating our work and the very positive reply.
> >
> > We wanted to share some final updates of the manuscript based on your suggestions.
> >
> > > **Weakness 4**: In experiment parts: a)Baselines like TCL should be compared, b) By identifiability, some metrics like MCC should be compared even though they are not component-wise identifiable.
> >
> > We have added two more chapters to our appendix:
> >
> > - Appendix L: Detailed discussion on **Component-Wise vs Linear Identifiability accompanying the MCC result table** [we posted in our previous reply](https://openreview.net/forum?id=ONfWFluZBI&noteId=wdZfaFoBJU).
> > - Appendix M: We now carefully **compare to baselines like TCL, PCL and TDRL** as requested and share [our codebase](https://anonymous.4open.science/r/dyncl-tdrl-baselines) building on [TDRL](https://github.com/weirayao/tdrl) which was proposed as a base also by another reviewer. We also use this opportunity to stress the differences in the theoretical foundations of these vs. our algorithm again. On dynamics inference, consistent with our theory, DynCL outperforms these added baselines.
> >
> > If there are any concerns remaining, please let us know, so we can address them in the remaining discussion period.

---

### Official Review · Reviewer_4X1L · 2024-11-03

**Soundness:** 3
**Presentation:** 3
**Contribution:** 2
**Rating:** 6
**Confidence:** 3

**Summary:**

In this paper, the authors propose a system-identification scheme for non-linear observations of non-linear time series data. In particular, they propose a modified contrastive learning set-up that posits linear latent dynamics. Compared to prior works in (time) contrastive learning, this directly enforces a notion of sequential temporal consistency, and seems to provide some benefit in system identification settings. Some supporting theory is provided, demonstrating that if the underlying dynamics are linear and invertible, then the proposed method asymptotically recovers the true dynamics up to affine ambiguity. For general non-linear systems, a (soft) switched-linear system heuristic is proposed, where Jacobian linearizations are applied at user-provided reference points.

**Strengths:**

Automatic identification of latent variables or dynamics is of critical importance in modern machine / reinforcement learning. The method the authors propose follows a line of self-supervised methods in contrastive learning. In comparison to its closest relative in time-contrastive learning (Hyvarinen and Morioka, 2016), the proposed method is seemingly more well-fit for fitting non-linear time-series data by fitting a latent time-series, rather than predicting a categorical label as in the aforementioned paper.

Since a main inductive bias built into the base method is that the latent dynamics are linear, the proposed method of iterative Jacobian linearizations is a sensible adaptation, and seems to benefit performance significantly.

Numerically, the proposed method appears to make contrastive methods more robustly performant.

**Weaknesses:**

In my opinion, the paper leaves quite a few critical questions unanswered, and in general suffers from a lack of polish. In its current state, I cannot recommend the paper for acceptance. The main weaknesses in my eyes are the following:

The paper claims to perform latent nonlinear system identification. This is a key desideratum in various fields such as reinforcement learning and continuous control, and thus has a rich history and literature. However, the assumptions in this paper--and notably how these inductive biases propagate to the algorithm design--severely restrict the applicability of the method without further evidence. Notably, a design assumption in this paper is that the observer function (i.e."mixing function") is invertible. This is a very strong assumption in the context of *non-linear system identification*, where even the foundational theory of linear system identification does not presume: in the Linear-Quadratic Gaussian (LQG) model, where the underlying state evolves linearly $x_{t+1} = Ax_t + Bu_t + w_t$, and observations are a linear function of state $y_t = Cx_t + v_t$ (ignoring the control input term for simplicity), the classical set-up has $d_y < d_x$, such that the observations are per-timestep a low-dimensional measurement of the underlying state. This immediately rules out the mixing function $g(x) = Cx$ being invertible, and this is precisely the motivation for notions such as observability/detectability. Partial observability presents the key challenge in non-linear sysID or reinforcement learning. In particular, it is well-known in controls and RL that ignoring partial observability and imposing a Markovian model (which this paper does implicitly by enforcing the state estimate as a function solely of the current observation) can lead to very undesirable outcomes. In the contrastive learning literature, partial observability is usually not a central issue, often because it is irrelevant for the motivating application (e.g. in computer vision), but one must address this problem for time-series data. In fact, the cited Time-Contrastive Learning method (Hyvarinen and Tomioka, 2016), despite making the same assumption in theory, actually propose a method that is more amenable to partial observability, since they predict categorical labels to *chunks* of observed data.

Regarding the polish of the paper, there are various typos and lacking definitions that make the paper hard to parse at times. The minor ones that I have caught are listed below. A particularly confusing point is the role of the control input $u_t$. The paper presents the control input as entering the latent dynamics directly. However, it is typically the case that the control input enters the state through a (possibly state-dependent) actuation matrix $\mathbf B(x_t) u_t$. In any case, how the control input enters the dynamics should be dependent on the parameterization of the dynamics, e.g. the affine ambiguity in $\mathbf L$ in the paper, which is not reflected in the authors' method as far as I can tell. Furthermore, it is unclear if the control input is available to the learner (which is usually the case in sysID), or if it is playing the role of stochastic noise, which eq (9) seems to suggest compared to eq (1). In either case, what role is the control input playing here: in the authors' set-up, there is no need to learn the actuation matrix, and the experiments involve learning a low-noise, nearly deterministic Lorenz system, which rules out some persistency of excitation effect (Tsiamis and Pappas, 2019).

**Minor comments/typos:**

Figure 1: x -> $x$

Page 3: "linear identifiability (...)", missing eqref?

Theorem 1: "bijective dynamics model $\mathbf f$", should probably mathematically define what that means.

Theorem 1: $\lambda$ is not defined in the main paper, only in the appendix.

Corollary 1: "$\hat{\mathbf f} :=1$", seems to be bad notation.

Beginning of Sec 4: "non-lineary" -> "non-linearly"

Equation (7): where is $g_k$ defined? Possible hash collision with mixing function notation.

Table 1: should probably introduce acronym "LDS" = Linear Dynamical System somewhere

Table 1: What does LDS$\downarrow$ mean?

Table 1: What do $\mathbf L$, $\mathbf L'$, $(\checkmark)$, $\mathbf I$ in the theory column indicate?

Between (11) and (12): "tailor" -> "Taylor"

Before Sec 5: "matices" -> "matrices"

Implementation paragraph: possibly missing number of A100 cards?

Eq (23): where is $p_u$ defined?

Eq (25): what does $p_{D}(y)$ denote precisely?

After Eq (50): "which is probably still fine because $\exp(-\|\mathbf L \cdot \|^2)$ is a valid kernel function (?)". This probably needs to be formalized/reworded.

**References:**

Anastasios Tsiamis and George J. Pappas, "Finite Sample Analysis of Stochastic System Identification", 2019.

**Questions:**

My main questions can be summarized as follows:

1. What is the marginal utility of this method rather than various other latent nonlinear dynamics estimation methods, e.g. (Watter et al., 2015) (which in fact also imposes locally linear latent dynamics), which do not make strong assumptions on identifiability?

2. How does enforcing the identifiability/invertibility conditions in this paper affect the method's performance in partially observed settings? This could be as simple as the LQG setting detailed above. Does this strong inductive bias translate to large errors when it is not satisfied (which is typically the case when only provided with observations of a ground-truth belief/latent state)?

3. As detailed above, what is the role of the control input? What is the effective difference of the proposed setting and an autonomous (latent) dynamical system $x_{t+1} = f(x_t) + \epsilon_t$?

4. From a practical perspective, how are the reference points for computing first-order linear approximation in the switching case chosen? Also, do these need to be recomputed per iteration, since the parameterization of $\hat{\mathbf{f}}$ changes per iteration?

**References:**

Watter et al., "Embed to Control: A Locally Linear Latent Dynamics Model for Control from Raw Images", 2015.

---

> ### Author Response · Authors · 2024-11-28
> **Final discussion and additions**
>
> Dear reviewer, thanks again for your extensive comments.
>
> We used the days since publishing the first version of our revision to provide even stronger results relating to the concerns you raised.
>
> Most importantly, we would like to point you to the additions we made to **Appendix I on handling non-injective mixing functions**, including the partially observed / LQS setting you proposed. We now added an even more data-efficient parameterisation of our model which achieves the same results with significantly less input data.
>
> Since you mentioned TCL as a method that is more amenable to partially observed data, we would like to refer you to our newest chapter on **additional benchmarking results of other time-series model baselines in Appendix M**. There we benchmark TCL (and other baselines like PCL and TDRL) against our SLDS setting. Even though TCL operates on (small) chunks of data and therefore could theoretically overcome partial observations, it is not able to model sufficiently complex dynamics and shows the worst performance in our additional experiments. This is most likely because TCL is designed with independently changing sources in mind. In contrast, our setting (Eq. 5) assumes latent dynamics. PCL performs quite well though (at ~90% vs. 99% R2 for DynCL), and might be the more appropriate baseline to compare to.
>
> **We very much hope that our combined revisions can resolve the concerns you have voiced in your review and would be happy to engage in further discussions in the remaining time.**
>
> In any case, thank you a lot for your time. Your input really helped improving the paper, and especially the time-lag formulation is a very practically relevant addition, in our opinion.

---

> > ### Comment · Reviewer_4X1L · 2024-11-30
> >
> > Thank you for the detailed reply and the Herculean effort put into the revision. After taking some time to re-read the paper after revision: 1. I believe all of my minor comments have been fixed 2. my main concerns regarding the relation to prior sysID works and ability to capture partially-observable systems have been addressed in good detail in the updated manuscript, 3. the proofs in the appendix have been fleshed out more convincingly. Accordingly, I have updated my score to leaning accept. I have also lowered my confidence by 1, since my remaining uncertainty comes from my own lack of knowledge of the frontier of contrastive learning, which I defer to the other reviewers, who seem to have more refined opinions therein.
> >
> > Thanks again to the authors for their hard work!

---

> > > ### Author Response · Authors · 2024-12-04
> > >
> > > Dear reviewer, thanks a lot for your final evaluation, it is great to hear that we managed to address your main concerns. We really appreciate the time you put into this review.

---

### Official Review · Reviewer_GCbH · 2024-11-04

**Soundness:** 3
**Presentation:** 3
**Contribution:** 3
**Rating:** 6
**Confidence:** 3

**Summary:**

The paper explores the use of self-supervised contrastive learning for dynamic system identification. It connects self-supervised learning (SSL) with identifiable representation learning, showing that SSL can identify system dynamics in latent space. The authors propose a model to uncover linear, switching linear, and non-linear dynamics under a non-linear observation model, providing theoretical guarantees and empirical validation.

**Strengths:**

1. The connection between contrastive learning and dynamic system identification is novel and could lead to simple encoder-only implementations favored in practice.
2. The synthetic experiments, especially the ablation studies, are extensive and investigate many aspects of the theoretical results.

**Weaknesses:**

1. The contributions (comparison with previous work), theoretical techniques, and novelty are very lightly discussed. I would appreciate a detailed discussion comparing this work's conditions with those in recent literature on temporal causal representation learning (e.g., [1] and its follow-up work). This would aid the contextualization of this work and make the contribution more transparent.

2. This work strikes me as theoretically oriented. There is a lack of discussion on the theoretical conditions, limitations, and implications. Just to name a few:

(a) the theorem requires the length for each time series to be infinite, which appears to be quite a strong assumption and not necessitated in recent work (e.g., [1], admittedly I am not an expert and would appreciate correction if I have misunderstood the statements). Why is this necessary, and would it be possible to show results for finite-length series?

(b) How does the control distribution influence the identification results? what conditions should it meet (does it have to be a time-independent Gaussian distribution)?

(c) Line 175: what is the significance of $A$ -- is it previously defined? Could you comment on lines 175-177?

3. Some minor issues:

(a) Line 145: what is (...)?

(b) Line 165: shouldn't $L^{\top} L$ be a matrix?


[1] Temporally Disentangled Representation Learning. Yao et al. NeurIPS 2022.

**Questions:**

See the weaknesses section.

---

### Official Review · Reviewer_KF3j · 2024-11-06

**Soundness:** 3
**Presentation:** 2
**Contribution:** 3
**Rating:** 6
**Confidence:** 2

**Summary:**

The main contribution of this paper is to provide identifiability theory for contrastive learning on time-series data with non-linear mixing, in the same spirit as time-contrastive learning (Hyvärinen & Morioka, 2016) for non-linear ICA. However, the authors discard the independence assumptions typically made in non-linear ICA with respect to the latent variables, and instead define a dynamical system as the data generating process. The proof operates under the assumption that the mapping from latent states to observables is injective but not necessarily linear, which is exploited to show that the composition of mixing and de-mixing by the model is an affine transform. As such, the estimated dynamics via contrastive estimation identify the true dynamics up to affine transformation in the latent space. There are experiments that corroborate the validity of this approach.

**Strengths:**

Applying contrastive learning to recover latent dynamics is itself a relatively new approach and the paper is well organised. The proofs use standard jacobian analysis tools and is easy to follow.

**Weaknesses:**

The paper needs refinement, with minor typos and inadequately captioned figures. While studying the identifiability of time-series contrastive learning might be novel, all the technical tools require carefully controlled assumptions and specific behavior of Jacobians under contrastive loss minimization, which typically do not hold in practice. Nonetheless, such assumptions are common in the literature on the identifiability of dynamical systems from observed time series.

**Questions:**

None

---

> ### Author Response · Authors · 2024-11-28
>
> Dear reviewer, thanks again for your feedback and the positive assessment of our work.
>
> With the final update of the manuscript, we added more discussion on our the assumptions (and their implications) of related work as part of our chapters on **"Component-Wise vs Linear Identifiability" (Appendix L)** and **"Comparison to Additional Time-Series Models" (Appendix M)** to further contextualise our results.
>
> Regarding your important question about **applicability of our assumptions in practice**, we want to point you again towards the reported improvements over CEBRA-Time (Schneider et al., Nature 2023) on real data for learning consistent latent dynamics (Appendix H, see "all-7"). The consistency metric reported can be regarded as a measure of empirical identifiability on real data in absence of knowing the true underlying distribution. The fact this metric reaches R2 scores around 90% (Figure 14b) points towards the real-world applicability of DynCL.
>
> Please let us know if we can address further questions you might have. If you have other questions that could increase your confidence or overall assessment of our work, please let us know, given that there is a full week of discussion time left. Thanks again!

---

### Author Response · Authors · 2024-11-28
**Final paper revision after the discussion phase**

We want to thank all reviewers again for their insightful comments which enabled us to substantially improve the paper during the rebuttal phase. **So far, three reviewers already acknowledged our efforts, unanimously positive.** Thank you very much for the positive reception of our work.

Given the substantial improvements to the paper during the first iteration and the tight reviewing timeline, **we understand that not all reviewers got the chance to reply yet**. We will stay available for additional discussions in the remaining week of the discussion period, and are **looking forward to hearing from 4X1L and KF3j**. We would also appreciate if **GCbH** and **3gxR** could have another look, since our new additions to the manuscript are touching on points raised by these reviewers. We post individual replies for more context.

To facilitate our final discussion, we added Appendix I, L, M to the manuscript to include a few more experimental results:

- Appendix I: We extended the results on **non-injective mixing functions** and show an even more data-efficient parametrisation of our model which improves results and supports the theoretical intuition on using embeddings based on multiple time-lags.
- Appendix L: We added **further discussion on component-wise vs linear identifiability** and are comparing our results w.r.t to the MCC metric and $R^2$ metric as requested by reviewer 3gxR.
- Appendix M: We added the promised **benchmarks on additional baselines**, namely TCL[1], PCL[2], and TDRL[3] as proposed by the reviewers GCbH and 3gxR. These benchmark results show the differences in theoretical assumptions of DynCL vs. established methods and demonstrate the performance of fitting dynamics model of DynCL. We provide the code (as close to [3] as possible) here: https://anonymous.4open.science/r/dyncl-tdrl-baselines.

**We are looking forward to engage in further discussion.** Thanks again!

---

References:
[1] Hyvarinen, A. and Morioka, H., 2017, April. Nonlinear ICA of temporally dependent stationary sources. In Artificial Intelligence and Statistics (pp. 460-469). PMLR.
[2] Hyvarinen, A., Morioka, H., 2016. Unsupervised Feature Extraction by Time-Contrastive Learning and Nonlinear ICA, in: Advances in Neural Information Processing Systems. Curran Associates, Inc.
[3] Temporally Disentangled Representation Learning. Yao et al. NeurIPS 2022.

Benchmarking codebase for Appendix M: https://anonymous.4open.science/r/dyncl-tdrl-baselines

---

### Author Response · Authors · 2024-12-04
**End of Rebuttal Summary**

**We would like to extend our sincere gratitude to the reviewers for their detailed feedback, constructive suggestions, and insightful discussions during the review process, and their unanimously positive final evaluation of our work.**

The changes we made to the manuscript are outlined in our two revision summaries [[1](https://openreview.net/forum?id=ONfWFluZBI&noteId=bg56TIwCs4), [2](https://openreview.net/forum?id=ONfWFluZBI&noteId=bg56TIwCs4)] as well as individual replies. The changes were also fully worked into the latest version of the manuscript and annotated accordingly.

As a result of our revisions and clarifications posted, reviewers GCbH and 3gxR raised their scores from 5 → 6, reviewer 4X1L from 3 → 6, and reviewer qeKY from 6 → 8, reviewer KF3j kept their initial positive assessment (6). **We thank all reviewers for the time they put into their initial evaluation of the paper, and the time into re-evaluating of our revised manuscript.** The reviews were very insightful and pushed us to do a range of important additions and controls. We believe that the added experiments, related work, and revisions towards clarity greatly enhance our initial submission, and are glad that the reviewers agree.

---

### Meta-Review · Area_Chair_ze57 · 2024-12-20

**Metareview:**

The paper studies representation learning for system identification of the time-series data. Namely, under the assumption that the ground-true time-evolution happens in the latent space, the authors examine the question of whether one can identify the system, i.e. learn the latent representation and the dynamics in the latent space from the time-series observations.

Building on the ideas of contrastive learning, the authors propose a model and prove a theoretical result for this model that in the limit of infinite time-series, the dynamical system can be identified up to a linear transformation (notably, assuming that the observation is obtained from the latent by an injective map). They apply their theory to identify the switching linear dynamics (linear dynamics in which parameters change in time by switching between a finite number of values). Furthermore, the authors argue that the non-linear dynamics can be approximated within the family of switching linear dynamics by the corresponding linearization of the dynamics around the time-series points in the latent space. Finally, the authors empirically validate the proposed framework for linear, switching linear, and Lorenz dynamical systems generating synthetic data in 3 and 6 dimensions.

Despite the reviewers' unanimous low confidence and the much room for improvement left in the presentation, I'm leaning toward accepting the paper. The main reason for this is that different reviewers found different parts of the work to be appealing. This signals that the paper might be relevant beyond a single community of researchers.

**Additional Comments On Reviewer Discussion:**

Most of the reviewers find the contributions of the paper interesting and valuable for the community. Namely,
(+) Reviewers KF3j, GCbH, and 3gxR highlighted the connection of the system identification with contrastive learning.
(+) Reviewers 4X1L, 3gxR, and qeKY acknowledged the importance of the result for time-series system identification.

The reviewers' major concerns are as follows:
(-) Poor presentation of the proposed method (Reviewer 4X1L, 3gxR, qeKY, GCbH, KF3j) -> improved by updating the manuscript.
(-) The main theoretical result assumes the injectivity of the map from the latent space to observables (Reviewer 4X1L) -> addressed by introducing time-lag embeddings in Appendix I and during the rebuttal.
(-) Empirical evaluation considers only synthetic data, and the baselines are not strong enough (Reviewer qeKY) -> addressed in the rebuttal and by providing additional results in Appendix H.

---

### Decision · Program_Chairs · 2025-01-22

Accept (Poster)